# Pandemic Restrictions in 2020 highlight the significance of non-road $NO_x$ sources in central London

Samuel J. Cliff[1], Will Drysdale[1], James D. Lee[1], Carole Helfter[2], Eiko Nemitz[2], Stefan Metzger[3, 4], and Janet F. Barlow[5]

[1]Wolfson Atmospheric Chemistry Laboratories, Department of Chemistry, University of York, York, YO10 5DD, UK
[2]UK Centre for Ecology and Hydrology, Bush Estate, Penicuik, EH26 0QB, UK
[3]Battelle, National Ecological Observatory Network. 1685 38th Street, Boulder, CO 80301, USA
[4]Dept of Atmospheric and Oceanic Sciences, University of Wisconsin-Madison, 1225 W Dayton St, Madison, WI 53711 USA
[5]Department of Meteorology, University of Reading, Reading, RG6 6BB, UK

**Correspondence:** Samuel J. Cliff (samcliff1@googlemail.com)

**Abstract.** Fluxes of nitrogen oxides ($NO_x$ = NO + $NO_2$) and carbon dioxide ($CO_2$) were measured using eddy covariance at the British Telecommunications (BT) Tower in central London during the coronavirus pandemic. Comparing fluxes to those measured in 2017 prior to the pandemic restrictions and the introduction of the Ultra-Low Emissions Zone (ULEZ) highlighted a 73 % reduction in $NO_x$ emissions between the two periods but only a 20 % reduction in $CO_2$ emissions and a 32 % reduction in traffic load. Use of a footprint model and the London Atmospheric Emissions Inventory (LAEI) identified transport and heat and power generation to be the two dominant sources of $NO_x$ and $CO_2$ but with significantly different relative contributions for each species. Application of external constraints on $NO_x$ and $CO_2$ emissions allowed the reductions in the different sources to be untangled identifying that transport $NO_x$ emissions had reduced by > 73 % since 2017. This was attributed in part to the success of air quality policy in central London, but crucially due to the substantial reduction in congestion that resulted from pandemic reduced mobility. Spatial mapping of the fluxes suggests that central London was dominated by point source heat and power generation emissions during the period of reduced mobility. This will have important implications on future air quality policy for $NO_2$ which until now, has been primarily focused on the emissions from diesel exhausts.

## 1 Introduction

Air pollution is thought to be the world's largest environmental risk to human health causing an estimated 7 million premature deaths every year (World Health Organisation, 2018). One species of pollutants of particular concern, especially in the UK, is $NO_x$. Formed as a by-product of high temperature combustion, $NO_x$ is commonly emitted from the tailpipe exhaust of internal combustion engine vehicles and through the use of fossil fuels to generate heat and energy in the residential, commercial and industrial sectors. The major component of $NO_x$ is nitrogen dioxide ($NO_2$); direct exposure to which is known to contribute to respiratory infections such as bronchitis and pneumonia (Ciencewicki and Jaspers, 2007). Indirectly, $NO_x$ is a key component to the photochemical formation of ozone and fine particulate matter ($PM_{2.5}$). Exposure to ozone and $PM_{2.5}$ has additional adverse effects on the respiratory and cardiovascular systems (Zhang et al., 2019). Consequentially, $NO_2$ and $PM_{2.5}$ were estimated

to cost the UK's National Health Service (NHS) and social care £1.6bn between 2017 and 2025, rising to £5.6bn if diseases with less robust evidence for an association are included (Public Health England, 2018). This has become particularly relevant since the start of the coronavirus pandemic whereby long-term exposure to air pollution has been associated with the severity
of COVID-19 cases (Imperial College London, 2021).

In 2008, countries in the EU were set legally binding limits for $NO_2$ concentrations in line with the World Health Organisation (WHO) recommendations. These are 40 $\mu g\ m^{-3}$ for the annual mean with no more than 18 exceedances of the 200 $\mu g\ m^{-3}$ hourly limit every year (European Parliament, 2008). This target was expected to be met by 2010. In 2021, the WHO reduced the recommended annual mean limit by 75 % to 10 $\mu g\ m^{-3}$ (World Health Organisation, 2021).

London is a megacity in the UK with extensive $NO_2$ air quality issues. Almost all roadside locations exceeded the European Limit Value for $NO_2$ every year between 2010 and 2016 (Font et al., 2019). Being in a highly developed position with significant resources, it has acted as a testing bed for policy intervention to try and curb emissions and achieve these air quality targets. These have been focused largely on traffic pollution and congestion charging and have the primary goal of reducing $NO_x$ concentrations via reduced road transport emissions, either through reduced traffic numbers or through reduced average
emission per vehicle per unit distance. Most notable is the introduction of the world's first ultra-low emissions zone (ULEZ), launched on 8th April 2019 with the zone spatially shown in Figure 1 a). This operates 24 hours a day, 364 days a year (excludes Christmas Day) and requires a daily payment if the vehicle driven inside the zone does not meet the most stringent emissions standards (currently Euro III for motorbikes, Euro IV for petrol cars and Euro VI for diesel cars and larger vehicles), in addition to the congestion charging payment within the same area. The ULEZ was expanded on 25th October 2021 up to
the north and south circular roads in an 18-fold increase in size. In addition to policy, the coronavirus pandemic had significant implications on $NO_x$ emissions in the UK through reduced mobility. During 2020 and 2021 the UK staged three lockdowns with "stay at home" orders. Full details on the timings and the severity of lockdown restrictions in London can be found in Figure A1.

Assessment of the impact of policy intervention and other external stimuli like the coronavirus pandemic on $NO_x$ emissions
is crucial for the future design and implementation of air quality policy in the UK. Eddy-covariance is a technique used to quantify the surface-atmosphere exchange of an atmospheric pollutant. The calculated flux coupled with a footprint model provides information on surface emissions, allowing for changes to be studied and for direct comparison to the emissions inventories used in policy development. Whilst most frequently used for measuring carbon dioxide exchange with ecosystems from stationary towers (Baldocchi et al., 2001; Griffis et al., 2008; Butterbach-Bahl et al., 2013), the technique has been
extended to the urban canopy for both greenhouse gases and air pollutants (Langford et al., 2010a; Lee et al., 2014; Helfter et al., 2016; Karl et al., 2017), as well as to airborne measurements for the assessment of fluxes at a much greater spatial extent (Vaughan et al., 2021; Metzger et al., 2013; Vaughan et al., 2017). Recently, Lamprecht et al. (2021) have used long term air pollutant emissions measurements to understand how COVID-19 restrictions have impacted different sources of $NO_x$ in the small European city of Innsbruck, Austria. We undertake a similar analysis, but for the megacity of London. This offers the
perspective of a different location where not only are emissions much higher, but contributions from different sources can vary significantly due to the nature of the activity required to support both greater population size and density. With the number of

megacities consistently increasing, and expected to reach 43 in 2030 (up from 31 in 2018), improving our understanding of the air pollutant sources in them is as critical as ever (United Nations, 2019).

Here we present the first year of data from the long-term $NO_x$ flux measurement programme at the BT Tower (London, UK). As the only long-term measurements of $NO_x$ emissions from a megacity in the world this is a highly unique and potentially informative data-set. The $NO_x$ emissions measurements are combined with additional $CO_2$ emissions measurements, a footprint model and the London Atmospheric Emissions Inventory (LAEI), and compared to previous measurements in 2017 to source apportion changes in emissions due to the coronavirus pandemic and the ULEZ. Resulting policy implications are inferred.

## 2  Experimental

### 2.1  Measurement Site

Instruments for measuring fluxes of urban air pollutants and greenhouse gases are situated in a small lab atop the BT Tower located in central London, UK (51°31'17.4"N, 0°8'20.04"W). The measurement height is 190 m above street level, with a mean building height of $8.8 \pm 3.0$ m in the 10 km radius surrounding the tower (Lee et al., 2014). The gas inlet and ultrasonic anemometer are attached to a solid mast that extends 3 m above the top of the tower. Air is pumped down a 45 m Teflon tube (3/8" OD) in a turbulent flow of 20-25 L $min^{-1}$ to the gas instruments, which are situated in a small air conditioned room inside the tower on the 35th floor.

### 2.2  $NO_x$ measurements

Long-term measurements of NO and $NO_2$ fluxes began in September 2020 with data presented here up to September 2021. Data is compared to previous measurements made by Drysdale et al. (2022) from March-August 2017. Both chemical species were measured using a dual channel chemiluminescence analyser (Air Quality Design Inc., Boulder Colorado, USA; 5 Hz) as described previously by Squires et al. (2020); Drysdale et al. (2022). The number of photons measured by the photomultiplier tube was converted into a part per trillion (ppt) mixing ratio using a five point calibration curve produced through dilutions of a 5 ppm NO in $N_2$ calibration standard (BOC Ltd., UK; traceable to the scale of the UK National Physical Laboratory, NPL) into $NO_x$ free air (generated from an external Sofnofil and activated charcoal trap). $NO_2$ was calculated by conversion of $NO_2$ into NO using a photolytic blue light converter (BLC). Here, both NO and $NO_2$ were measured, from which $NO_2$ can be quantified by subtracting the NO mixing ratio and applying a correction factor for the conversion efficiency of the BLC. The instrument was calibrated every 37 hours in addition to an hourly zero measurement to subtract the temperature dependent background signal of each channel. The uncertainty of the NO measurement is given as $\pm$ 3%, resulting from uncertainties in the sample mass flow controller, calibration gas mass flow controller and calibration gas certification. The uncertainty for the $NO_2$ measurement is given as $\pm$ 4.7% due to the additional uncertainty in the conversion efficiency calculation, determined in the laboratory via variation in repeated tests. The precision for each channel is calculated as 53 ppt and 184 ppt for NO and $NO_2$ respectively from the standard deviation in all the hourly zeros during the measurement period.

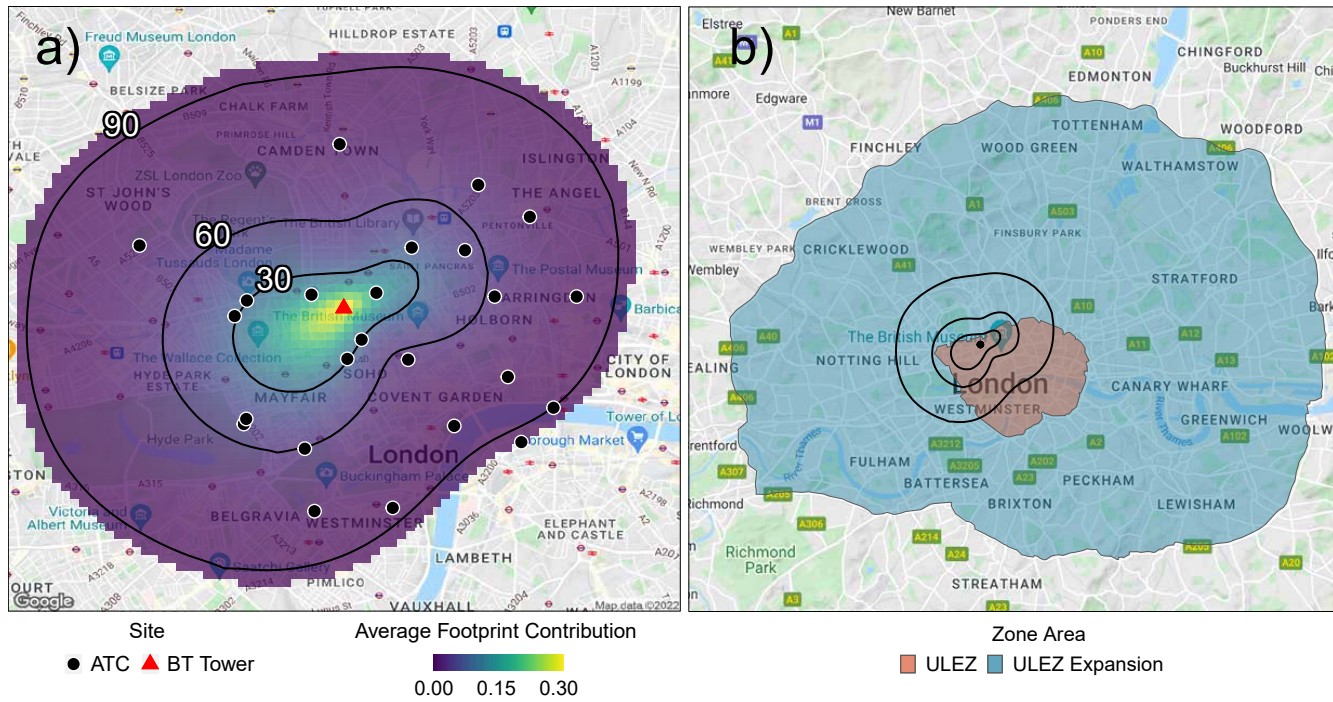

**Figure 1.** a) The average footprint climatology for the Sept 2020-Sept 2021 time period, with the 30, 60 and 90 % contribution contours and the location of the 24 ATC sites and the BT Tower site overlaid. b) Spatial boundary of the ULEZ (red) and ULEZ expansion (blue) with the same footprint contribution contours presented in a) overlaid as black lines, and the BT Tower site marked as a black dot. Maps produced from Google Maps (© Google Maps 2022) accessed using an API in R.

### 2.3 CO$_2$ measurements

Long-term measurements of CO$_2$ have been ongoing at the BT Tower since 2011 as part of UKCEH's National Capability programme. Dry mass fractions ($1\sigma$ precision $< 300$ ppb) were measured initially using a cavity ringdown spectrometer (Model 1301-f, Picarro Inc., Santa Clara, California, USA; 10 Hz) as described by Helfter et al. (2016). Unfortunately, instrumental failure means data is not available between February and June 2021, after which a closed path infrared gas analyser (Li-7000, LI-COR Environmental, Lincoln, Nebraska, USA; 10 Hz) took over the CO$_2$ flux measurement.

### 2.4 Meteorological measurements

Meteorological measurements were made at the BT Tower as described by Lane et al. (2013). Wind speed, wind direction and sonic temperature were measured using a ultrasonic anemometer (Gill R3-50, Gill Instruments, Lymington, UK; 20 Hz), along with pressure and relative humidity measurements using a weather station (WXT520, Vaisala Corp. Helsinki, Finland; 1 Hz). For ease of processing, each chemical species is logged separately at it's maximum measurement frequency into a file with the sonic anemometer data averaged to the same measurement frequency.

## 2.5 Flux Calculations

The flux, F, is defined in this context as the vertical transport of a chemical species per unit area per unit time. Hourly fluxes were calculated using eddy covariance theory as described by Eq. (1), where F is equal to the covariance between the instantaneous change in vertical wind speed, $w'$, and the instantaneous change in species concentration, $c'$, averaged over the hour.

$$F = \overline{w'c'} \tag{1}$$

Eddy-covariance calculations were performed using the modular software packages in eddy4R adopting the same processing settings described in Drysdale et al. (2022) as adapted from Squires et al. (2020) This was to allow a direct comparison to be made to the previous measurements made in 2017. The lag time correction was determined by maximisation of the cross-covariance between the pollutant concentration and the vertical wind component with an additional application of a high-pass filter which improves the precision of the determined lag time by an order of magnitude (Hartmann et al., 2018; Squires et al., 2020). This resulted in median lag times of 7.2 s for NO, 7.6 s for $NO_2$ and 21 s for $CO_2$. Data was filtered such that the friction velocity (u*) is $> 0.2$ to ensure sufficiently developed turbulence and using eddy4R's quality control flagging scheme. The QA/QC process is described in detail by Smith and Metzger (2013); Metzger et al. (2022). Data is flagged as either valid or invalid based on the combination of individual flags for input data validation, homogeneity and stationarity, and development of turbulence.

## 2.6 Flux uncertainties

### 2.6.1 $NO_x$ chemistry

Eddy covariance has traditionally only been used for relatively unreactive greenhouse gases like $CO_2$ with long atmospheric lifetimes. Attempting the calculation of $NO_x$ fluxes is potentially problematic due to the greater reactivity and hence shorter lifetime of the species. If the loss rates of the reactive species is of a similar timescale to the vertical transport to the measurement height, the measured flux would be an underestimate and would not be representative of those emitted at the ground. In the case of $NO_x$, the major loss route to the atmosphere is via the reaction between $NO_2$ and OH. The rate constant for this simple association reaction can be calculated for the BT Tower specific conditions from Eq. 2 using mean values of temperature (T, 289 K) and pressure (P, 989 hPa) (Jet Propulsion Laboratory, 2020). This is derived from the low-pressure limiting rate constant ($k_0(T)$) and the high-pressure limiting rate constant ($k_\infty(T)$) using location specific total gas concentrations ([M]). n and m are simple exponents for the given reaction, in this case 3 and 0 respectively.

$$k_f(T, [M]) = \left\{ \frac{k_\infty(T) k_0(T) [M]}{k_\infty(T) + k_0(T) [M]} \right\} 0.6^{\left\{ 1 + \left[ log_{10} \left( \frac{k_0(T)[M]}{k_\infty(T)} \right) \right]^2 \right\}^{-1}} \tag{2}$$

Where:

$$k_0(T) = k_0^{298} \left( \frac{298}{T} \right)^n \tag{3}$$

$$k_\infty(T) = k_\infty^{298} \left( \frac{298}{T} \right)^m \tag{4}$$

$$[M] = \frac{PA_v}{RT} \tag{5}$$

Assuming a simple first order loss rate, the level of $NO_x$ loss to the atmosphere can then be estimated from Eq. 6 using the previously determined rate constant ($k_f(T, [M]) = 1.973 \times 10^{-11}$), the concentration of OH ([OH]) and the transport time to the measurement height (t).

$$\frac{[NO_x]}{[NO_x]_0} = e^{-k[OH]t} \tag{6}$$

Since [OH] is not routinely measured in London, a typical midday summers value $2 \times 10^6$ is used from London measurements in 2012 which would represent the maximum loss rate observed throughout the year (Lee et al., 2016). Barlow et al. (2011) estimate a typical transport time of < 10 minutes for the BT Tower, although under stable conditions this could increase to 20-50 minutes. This results in a loss of 2%, increasing up to 11% for a 50 minute transport time. The 11% loss represents the maximum loss observed at the BT Tower since it occurs under the most stable conditions during peak OH concentrations for London in Summer. In reality, this level of loss will not be observed in the data since stable conditions are filtered out in the QA/QC process and the majority of the OH concentration present throughout the year is less than that used in this calculation. Since it is much more likely to be at or below the 2% threshold, we consider $NO_x$ reactivity to be a minor uncertainty in the flux calculations and a correction is not applied.

### 2.6.2 Vertical flux divergence

Another source of uncertainty is the size of the measurement height relative to the boundary layer. At 191 m, the sample inlet and sonic anemometer is often an appreciable portion of the boundary layer and can extend above the constant flux layer. On occasion, this results in concentration enhancements below the measurement height and an underestimation of the surface flux through vertical flux divergence. The impact of storage and vertical flux divergence at the BT Tower has been discussed previously by Helfter et al. (2016); Drysdale et al. (2022), and in the absence of concentration and wind measurements at different heights up the tower, remains to be a notable source of uncertainty in the measurement. Helfter et al. (2016) speculates that venting after the onset of turbulence would capture some, if not most of the material stored below the measurement height. Drysdale et al. (2022) demonstrates a correction for vertical flux divergence as a function of effective measurement height and effective entrainment height. The correction was typically around 20% for 2017, but is not applied to the data due to uncertainties in the boundary layer height data. Since this work studies relative magnitudes between two periods, the impact of VFD will likely cancel out, provided the meteorology is similar. Boundary layer height was on average 8% lower in 2020/21 compared to the 2017 measurement period. As such, a comparison between the VFD correction presented in Drysdale et al.

(2022) for both periods was conducted. The lower boundary layer height in 2020/21 meant the correction was slightly higher at 24 %. A discussion on the impact this had on the results of this paper is given in Section 3.

### 2.6.3   High/low frequency loss

Due to the height of the measurement and the large eddy size above the urban roughness layer, the high frequency contributions to the fluxes are expected to be small. Drysdale et al. (2022) calculated high frequency loss for NO and $NO_2$ at the BT Tower via co-spectra relative to temperature measured at 20 Hz. Correction factors above 1 Hz were shown to be of the order of 2 - 3 %. Losses due to low frequency can occur due to an insufficient length of averaging period. Previous studies at the BT Tower for 30 minute flux averaging periods have calculated losses due to high-pass filtering to be $< 5\%$ (Helfter et al., 2011; Langford
et al., 2010b). Since a 60 minute averaging period is used here, loss will be even lower. Both of these errors are considered minor and as a result no correction has been applied.

### 2.7   Footprint Modelling

A parameterised version of the backwards Lagrangian stochastic particle dispersion model implemented in eddy4R was used to estimate the footprint for each hourly flux measurement at the BT Tower. The model is described by Kljun et al. (2004)
and has been parameterised for a range of meteorological conditions and receptor heights to reduce the computational expense of running it. The original model aims to produce a cross-wind integrated footprint function as a function of its along-wind distance, which has now been further extended into two dimensions using a Gaussian distribution driven by the standard deviation in the cross-wind component (Metzger et al., 2012; Kljun et al., 2015). Meteorology statistics from the eddy covariance calculations are used in combination with modelled boundary layer height from ERA5 (Copernicus Climate Change Service
Climate Data Store (CDS), 2021), and a surface roughness length of 1.1 m to produce a weighted matrix of 100 m x 100 m grid cells. Each output weighted matrix was then scaled and aligned to an appropriate coordinate reference system to allow each matrix to be plotted onto a map.

### 2.8   Traffic Data

Hourly traffic loads surrounding the BT Tower were calculated by summing the traffic load from each of the 24 Automatic
Traffic Counters (ATCs) within the flux footprint, as shown in Figure 1. This gave an indication of the magnitude of traffic load for both measurement periods and allowed relative changes to be studied between the two years. In addition, daily vehicle length breakdown was examined from which vehicles were separated into three length classes: $< 5.2$ m, indicating the number of passenger cars, 5.2 m-12 m, indicating the number of vans and rigid lorries and $> 12$ m, indicating the number of buses and articulated lorries. As the LAEI estimates that almost all lorry emissions in central London are due to the rigid class, the $>12$
185  m class is assumed to solely be made up from buses. Data was provided by the Operational Analysis Department, Transport for London (TFL) via a freedom of information request (Transport for London, 2021).

## 2.9 Emissions inventories

The London Atmospheric Emissions Inventory (LAEI) is a spatially disaggregated 1 km$^2$ gridded map of the annual emissions of various air pollutants for the London area up to the M25 motorway ring road (Greater London Authority, 2016). Annual emissions are estimated using emission factors and activity factors for the different sources. For example, emissions from domestic combustion in (tonnes/year) would be calculated as Gas Consumption (GW.h/year) × Emission Factor (t poll/GW.h). The inventory is produced roughly every 3 years by TFL and the Greater London Authority (GLA). At the time of writing, there was no inventory estimates for the pandemic affected years of 2020 and 2021. We therefore use the most recent version produced in 2016, which relates to a 'normal' year unaffected by lockdowns, to understand how emissions have changed since the 2017 measurements.

## 3 Results and Discussion

Of the 8760 hours in the year, 7034 hours of $NO_x$ fluxes were calculated. Data loss was largely due to instrument or sample pump failure. Of these 7034 hours, a further 3621 were removed by the quality control flagging to leave 3413 hours of $NO_x$ fluxes to be analysed. This data is displayed in Figure 2 along with measured $CO_2$ flux, traffic load around the tower and the UK's restrictions stringency index as calculated by the Oxford COVID-19 Government Response Tracker (Hale et al., 2021). Traffic load around the tower was strongly anti-correlated with stringency index as expected. However, there was no obvious correlation of $NO_x$ flux with traffic load. In fact, $NO_x$ flux displays an anti-correlation with traffic flow and stringency index from April to August. This is likely due to a reduction in heat and power generation emissions due to the warmer weather, which is a first indication that traffic may not be the dominant source of $NO_x$ flux during this period. Traffic loads had not recovered to pre-pandemic levels either (Figure A3) despite the fact all lockdown restrictions were fully removed on 21 June 2021, hinting at a more long-term change in behaviour. This is not unexpected as the stringency index remained at 40 %, mainly due to self-isolation requirements and international travel restrictions which were still present at this stage. In the absense of a long term time series from which a number of studies have used boosted regression models to predict normal emission scenarios, we compare data to that previously measured from March - August 2017 by Drysdale et al. (2022). This is an ideal time period for comparison since the measurement footprint was very similar (see Fig. A2) and meteorological conditions meant minimal bias was expected between the years (see Sec. 2.6.2). Average diurnal $NO_x$ fluxes were down 73 % (3.45 vs 12.88 mg m$^{-2}$ h$^{-1}$). However, only a corresponding 20 % reduction in $CO_2$ flux (2455 vs 3062 mg m$^{-2}$ h$^{-1}$) and 32 % reduction in traffic load (16540 vs 24405 vehicles day$^{-1}$) around the measurement site was observed. These changes can be clearly seen in the diurnal profiles in Fig. 2. These % changes were calculated after application of the different vertical flux divergence corrections discussed in Section 2.6.2, and exhibited a negligible variation of $< 1$ %.

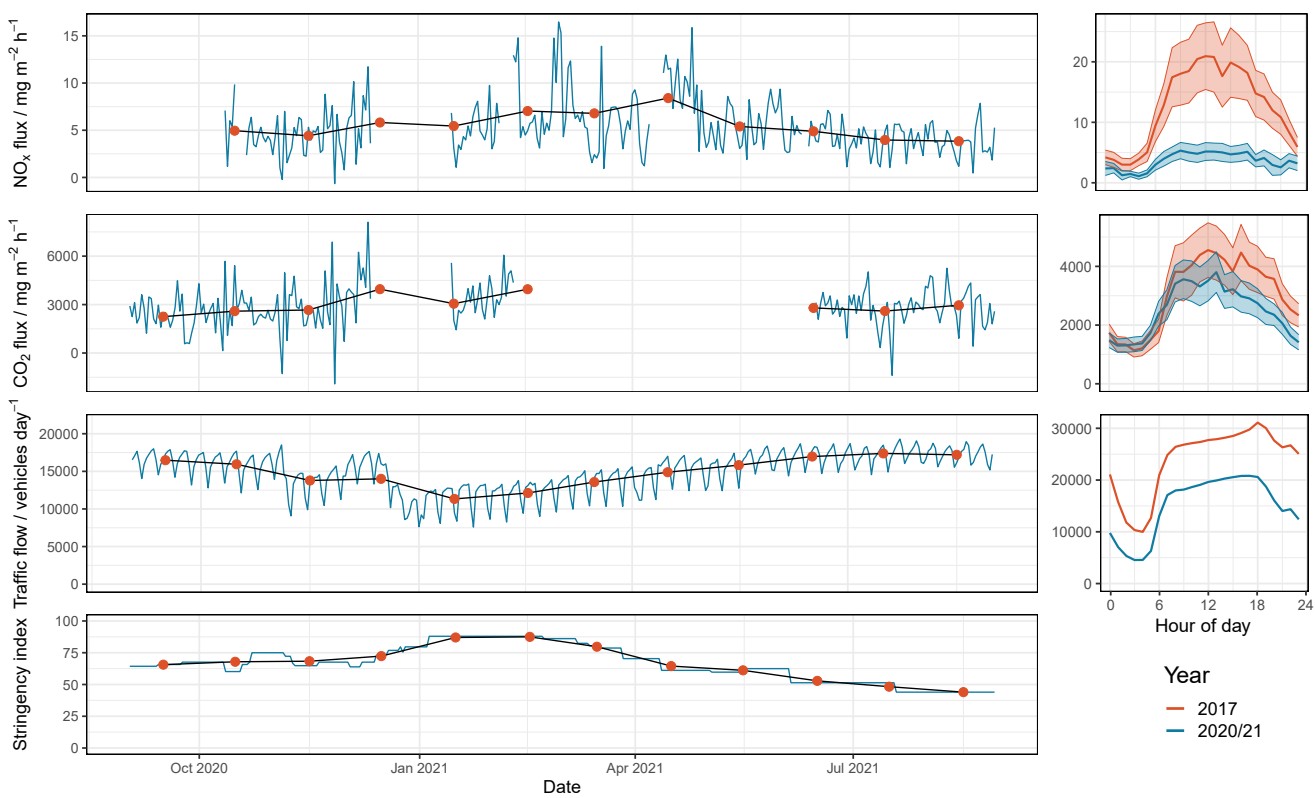

**Figure 2.** Time series from September 2020 to September 2021 for daily time averaged $NO_x$ flux, $CO_2$ flux, traffic load around the tower and stringency index in the UK. Also shown are monthly averages for each variable as red dots centered around the center of each month. Average median diurnal profiles with error bars (calculated as the combination of random and systematic errors in the flux calculations, as described by Mann and Lenschow (1994)) for the data are shown to the right in blue for 2020/21 in comparison to those generated from the 2017 data in red.

## 3.1 Calculation of inventory estimated emissions

Estimated emissions from the London Atmospheric Emissions Inventory (LAEI) for the measurement footprint were calculated to aid understanding of these observations. The hourly footprint weighted matrix output from eddy4R was used to select the relevant areas of the LAEI. The theoretical contribution to the flux was extracted from each footprint grid cell and scaled for hour of day, day of week and month of year for each emissions sector using a set of anthropogenic scaling factors described by Drysdale et al. (2022). An excellent agreement between the diurnal profiles and measurement footprint (shown in Figure A2) for the 2017 and 2020/21 measurement periods was seen. This gave us confidence that any changes in emissions were not to do with sampling in different times of year or sampling in different areas of central London. The source breakdown of the 2017 inventory generated time series for both $CO_2$ and $NO_x$ is shown in Figure 3. Emissions of both species are almost entirely

made up from combustion of fossil fuels to generate heat and power in the domestic, commercial and industrial sectors and the transport sector, which is dominated by various forms of road transport. However, each species has a significantly different relative contribution from each sector. 75 % of $CO_2$ emissions are estimated to arise from heat and power generation but only 42 % of $NO_x$ emissions from the same source.

### 3.2 Source apportionment of emissions reduction

The inventory breakdown for each species and the different percentage reductions in measured emissions since 2017 were used to disentangle changes in emissions of each sector. This was done simultaneously using a number of assumptions.

**Table 1.** A summary of the data used in the formation of simultaneous Eqs. (7) and (8).

| | $NO_x$ | $CO_2$ |
|---|---|---|
| % measured reduction in emissions since 2017 | 73 % | 20 % |
| % contribution from heat and power generation | 0-50 % ($\alpha$) | 75 % |
| % contribution from transport | 50-100 % ($\beta$) | 25 % |
| % reduction in heat and power generation emissions | $x$ | $x$ |
| % reduction in transport emissions | $y'$ | $y$ |

Helfter et al. (2016) highlight the excellent agreement of $CO_2$ emissions measured at the BT Tower to those estimated by the LAEI and thus the source contributions of 75 % from heat and power generation and 25 % from transport for $CO_2$ are taken as accurate here. On the other hand, previous observations have shown a significant underestimation of $NO_x$ emissions in
central London (Vaughan et al., 2021; Drysdale et al., 2022). This is most likely due to an underrepresentation of road transport $NO_x$ emissions in line with a poor representation of diesel vehicle emissions and/or congestion. Therefore, rather than using the inventory predicted 42:58 split for heat and power generation:transport the relative contributions were varied. Labelled as $\alpha : \beta$, different scenarios between 50:50 and 0:100 (where $\alpha + \beta = 100\%$) were chosen to represent all possible levels of $NO_x$ underestimation. The percentage reduction in heat and power generation emissions for both species is labelled as $x$ with the
assumption that any reduction in this sector's emissions would have the same reduction in measured flux for both species. With minimal legislation for the sector introduced between 2017 and 2020/21 and a failure to address $NO_x$ emissions from boilers in the UK's Clean Air Strategy, this assumption is considered reasonable (Department for Environment, Food and Rural Affairs (DEFRA), 2019). However, this is is likely to be untrue for transport. Policy implemented between the two measurement periods specifically targeted $NO_x$ emissions and $NO_x$ emissions are disproportionately higher in higher traffic loads due to the
ineffectiveness of exhaust treatment systems in that environment. Additionally, the modernisation of the vehicle fleet will have introduced more vehicles with lower $NO_x/CO_2$ emission ratios. Therefore, the relative change in the emissions of $NO_x$ and $CO_2$ from traffic sources may not have been the same, and different values are given here as y and y'. This information is all summarised in Table 1 with the two independent constraints displayed in Eq.'s (7) for $CO_2$ and (8) for $NO_x$:

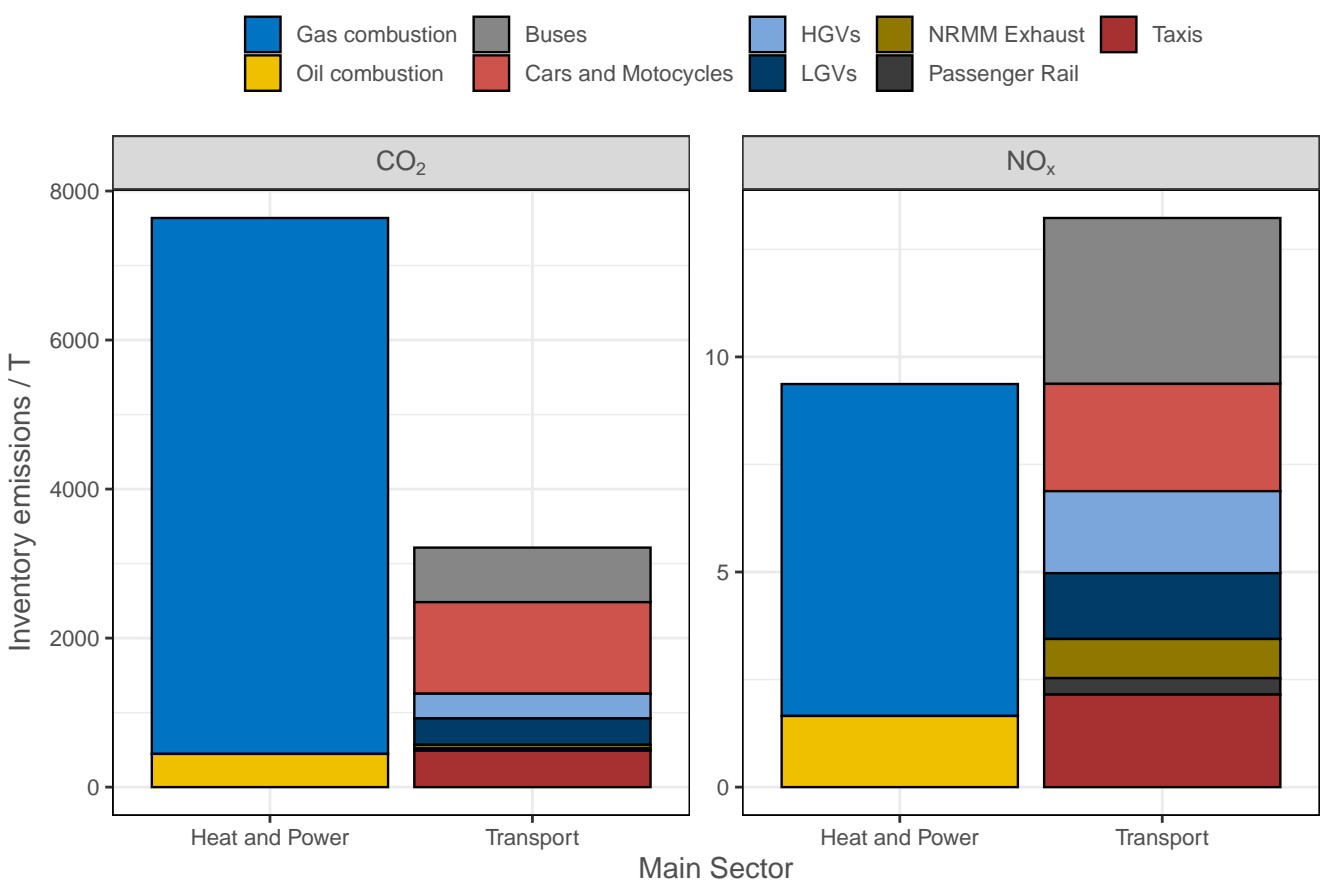

**Figure 3.** Inventory estimated breakdown of emissions for $CO_2$ (left) and $NO_x$ (right) in Tonnes for March through August of 2017 as determined from the inventory emissions extraction for our hourly measurement footprint. LGV = Light Goods Vehicles, HGV = Heavy Goods Vehicles, NRMM = Non-Road Mobile Machinery.

$$\delta CO_2 = 0.75x + 0.25y_{[32,100]} = 20\% \tag{7}$$

$$\delta NO_x = \alpha x + \beta y'_{[73,100]} = 73\% \tag{8}$$

The different scenarios are visualised in Figure 4 to aid understanding of the possible solutions. Two bounding conditions drawn as hashed lines are applied to constrain the solutions. These are as follows: a) A reduction in transport emissions greater than 100 % is not possible and b) $CO_2$ emissions from transport must have decreased by at least 32 % in line with the 32 %

reduction in traffic load. In reality, $CO_2$ emissions will have decreased by greater that 32% as a result of the fleet modernisation

which has lead to a decrease in the average $CO_2$ emissions per new vehicle registration (European Environment Agency, 2022).

     Highlighted in green are all the resulting possible solutions where crucially, to achieve the observed reductions in $NO_x$ flux, there must have been a 73-100 % reduction in transport $NO_x$ emissions with transport contributing $> 70$ % to total $NO_x$ emissions. This transport contribution percentage demonstrates the underestimation in the inventory of transport emissions in agreement with Karl et al. (2017) and Drysdale et al. (2022). However, the most interesting observation is that a 73-100 %

decrease in transport $NO_x$ emissions is seen for only a 32 % decrease in traffic load since 2017.

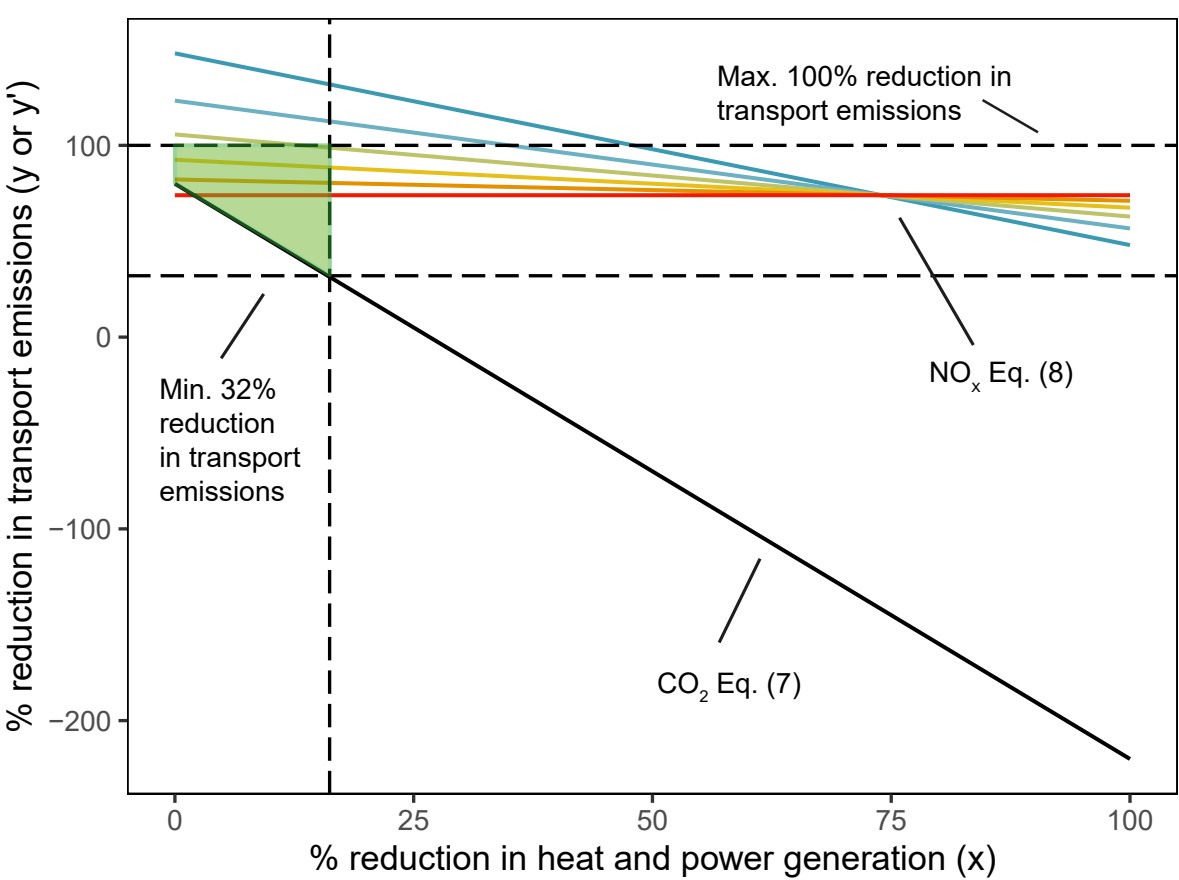

**Figure 4.** A plot showing the external constraints on $NO_x$ and $CO_2$ emissions. Eq. (7) is shown as the solid black line and Eq. (8) as the coloured solid lines, with each colour reflecting a different value of $\beta$. The two horizontal hashed lines represent the two discussed constraints, and the vertical hashed line shows the constraint the minimum of 32 % reduction in traffic $CO_2$ emissions has on the $NO_x$ scenarios. All possible solutions when the constraints are applied are highlighted by the shaded green area.

When compared to concentrations, we found that $NO_x$ concentrations at Marylebone Road, a kerbside monitoring site within the flux footprint, had declined by 62 % between the two periods (248 vs 95 $\mu$g m$^{-3}$). This is increased to 69 % (214 vs 67 $\mu$g m$^{-3}$) when looking at the roadside increment concentration (roadside – urban background) as determined from Marylebone Road and London North Kensington monitoring sites. Whilst this was slightly lower in magnitude than the measured change in flux, the concentration data will have been heavily influenced by meteorology and so some disagreement was expected. There are a number of plausible explanations for the large decrease in $NO_x$ fluxes. The introduction of the ULEZ is thought to have resulted in a pre-pandemic 31 % reduction in $NO_x$ emissions from road transport in central London (Greater London Authority, 2020); this is likely to be an upper estimate for our measurements due to the fact a significant proportion of our flux footprint being situated outside of the ULEZ zone. With average traffic loads between April and November 2019 after the ULEZ was introduced only down 1.8 % on 2018 levels for the same period, the vast majority of the reduction is due to a clean-up of the fleet which reduces the emissions per vehicle per unit distance. The remaining emissions are further reduced by 32 % due there being 32 % less vehicles on the roads surrounding the BT tower during the pandemic. This leaves   20-45 % of unaccounted for emissions reduction. A small portion of this unaccounted for reduction may be due to a 40 % reduction in the number of buses on the roads surrounding the BT Tower during 2020/21. The >12 m class of the vehicle length breakdown in Figure A4 represents the bus classification. With buses making up 17 % of the total $NO_x$ emissions from road transport, the decrease in relative proportion between 2017 and 2020/21 could result in a maximum of 7 % extra reduction in $NO_x$ emissions. It is likely to be less than 7 % due to the small increase in the relative proportion of the 5.2 m-12 m class.

### 3.3   Flux correlations with traffic load

Examining how the $NO_x$ flux correlated with traffic load for both measurement time periods gives further insight into the unaccounted emissions reduction. Figure 5 generally shows significantly enhanced $NO_x$ emissions in 2017 above 25000 vehicles hr$^{-1}$. With road transport being the dominant source during these measurements it is highlighting what is thought to be the effect of congestion on $NO_x$ emissions.

During periods of high congestion, increased emissions are expected due to increased length in journey time, a greater number of accelerations in the stop start nature of traffic and reduced effectiveness of exhaust gas $NO_x$ treatment systems in diesel vehicles at low engine temperatures (Carslaw and Rhys-Tyler, 2013). The effect of congestion on $NO_x$ emissions is highly dependent on several variables including the fleet composition, type of exhaust treatment system and the actual level of congestion (Ko et al., 2019). It is thought that for individual roads, excess emissions from congestion can be anything up to 75 % greater than non-congested roads (Gately et al., 2017). Therefore, it is thought that reducing the peak traffic load below 25000 vehicles hr$^{-1}$ has had a large impact on traffic $NO_x$ emissions, more than accounting for the remaining emissions reduction.

### 3.4   Spatial Mapping

This change in emissions is clearly seen in the spatial mapping of the $NO_x$ fluxes in Figure 6. Drysdale et al. (2022) assigned the heightened emissions to the northeast of the BT Tower in Figure 6 a) to Euston station, including not only the train station,

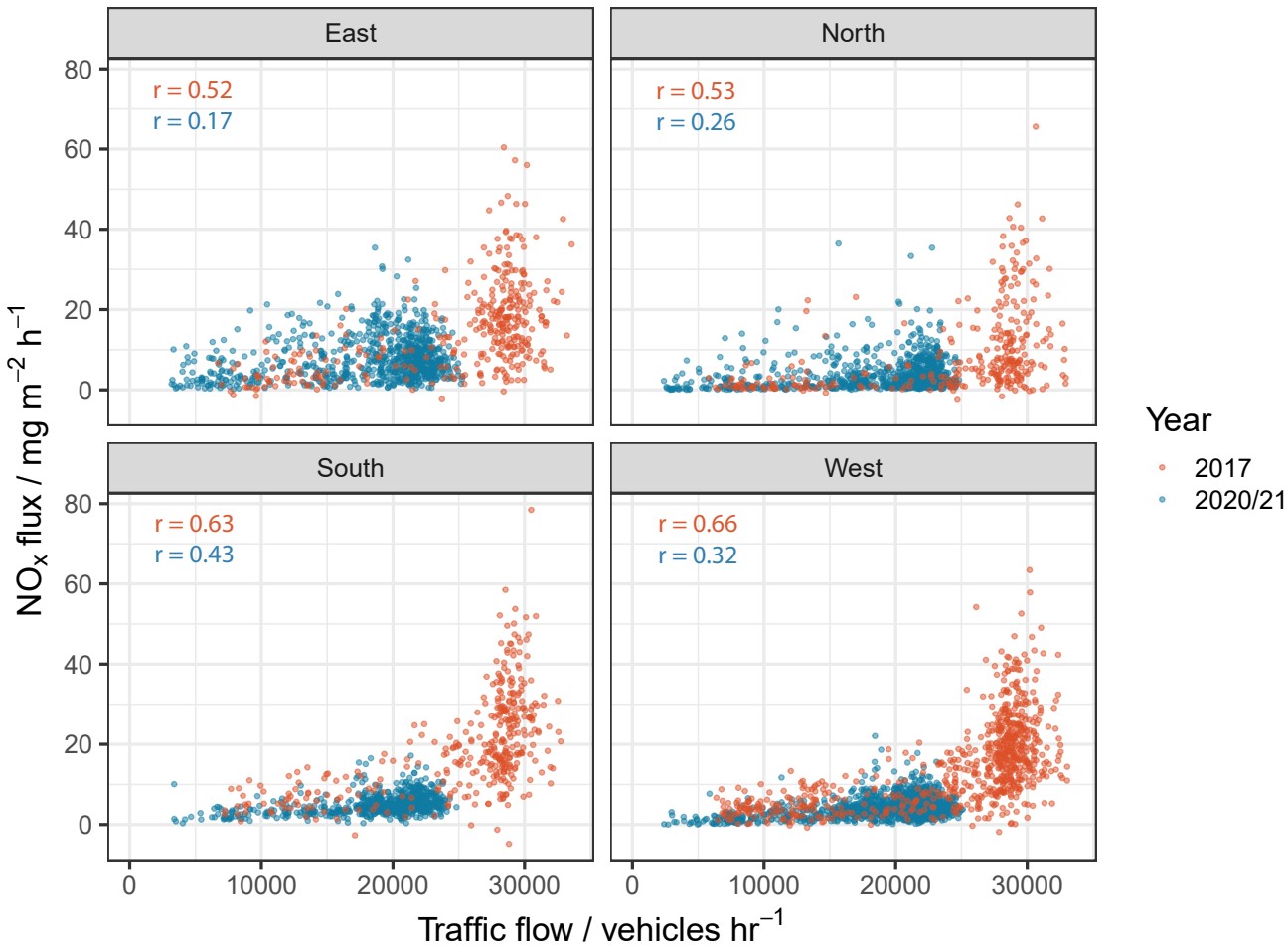

**Figure 5.** Comparison of the measured $NO_x$ flux with hourly traffic load (sum of the 24 surrounding ATCs) for March through August 2017 (red) and Sept 2020 - Sept 2021 (blue). The data is split by wind direction: North ($315°$-$45°$), east ($45°$-$135°$), south ($135°$-$225°$) and west ($225°$-$315°$). In the top left of each facet is the Spearman correlation coefficient for each year for the corresponding wind direction.

but also the large Euston bus station, taxi ranks and busy roads feeding the station. In addition, the high fluxes measured to the southwest are assigned to highly congested streets such as Oxford Street, Regent Street and Piccadilly. Both areas are associated with high traffic volumes and congestion and support the notion that road transport emissions dominate in central London in 2017. However, these areas have almost an order of magnitude smaller emissions and are barely visible for 2020/21 when shown on the same scale. This adds additional support to the conclusion that reduced traffic load and thus congestion in 2020/21 have been a major cause of the reduced $NO_x$ flux.

The spatial map for 2020/21 in Figure 6 c) on its own scale identified a shift in the dominant $NO_x$ emissions source between 2017 and 2020/21. Whilst Euston station and the previously congested southwesterly area are still noticeable, the major stand out area is to the east. Depicted as a white box is the outline of the University of London point source as documented by the National Atmospheric Emissions Inventory, a similar inventory to the LAEI but for the whole of the UK (Defra and BEIS). The University of London is the largest university in the UK and its Bloomsbury Campus appears directly under the heightened emissions area. This site is made up of much of the University College London (UCL) central administration, the UCL hospital and Bloomsbury Heat and Power, a number of combined heat and power (CHP) sites to power the university. CHP systems simultaneously generate heat and electrical power from a single source of energy. By capturing and utilising the heat that is generated as a by product of the electricity generation process, efficiency is increased which can reduce carbon emissions by up to 30 % compared to conventional separate generation (Department for Business, Energy and Industrial Strategy (BEIS), 2021). However, the requirement for CHP to be in urban areas risks an increase in air pollution. Indeed it has been shown that CHP can "substantially" impact air quality due to $NO_x$, the highest criteria judged by Environment Protection UK and the Institute of Air Quality Management (Kings College London, 2018). Here, the heat and power generation source stands out and dominates over transport but is only seen due to the drastic reduction in transport $NO_x$ emissions during the period of pandemic reduced mobility. The Spearman Correlation coefficients presented in Figure 5 give further evidence that the dominant source between the two periods has changed. Correlations between $NO_x$ flux and traffic load are reduced in 2020/21, in particular in the Easterly direction. Here, the lowest correlation is observed and high $NO_x$ fluxes are seen even at low traffic loads. These observations are in agreement with the spatial mapping interpretation in that heat and power generation is the dominant source from this direction.

## 4  Conclusions

Eddy covariance emissions measurements at the unique BT Tower site in central London provide an opportunity to study the evolution of air pollutant emissions in a megacity and the part that policy and other external stimuli play in improving air quality. Here, the direct emissions measurements have shown that reducing congestion could be an even more effective way of reducing $NO_x$ emissions from road transport than the ULEZ. However, this is not the direction in which the UK is heading. With much cheaper mileage, the continued uptake of electric vehicles is predicted to increase congestion. Reducing the number of vehicles on the road by improving infrastructure for other greener methods of travel such as cycling would not only achieve reduced congestion but give additional benefits to health further reducing costs of treatment at health services (Fishman et al.,

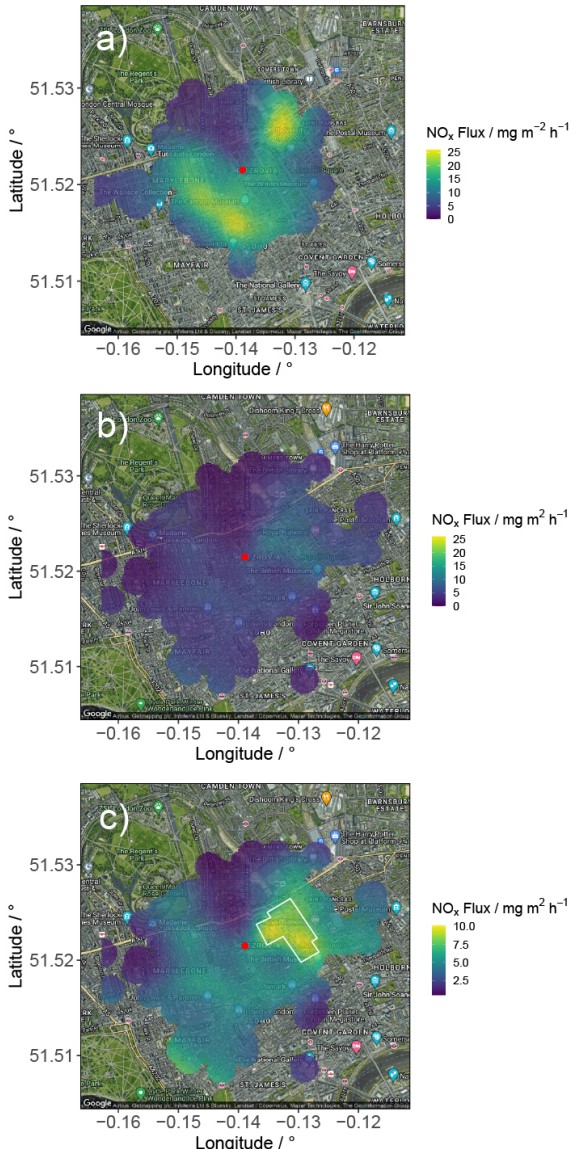

**Figure 6.** NO$_x$ flux surfaces as a function of along-wind distance to the footprint maximum contribution and wind direction, as derived by Drysdale et al. (2022), for a) March through August 2017 and b) September 2020 to September 2021, displayed on the same scale for ease of visual comparison. Also shown in c) September 2020 to September 2021 on it's own scale with the border for the University of London overlaid in white. The location of the BT Tower site is displayed as a red dot in each spatial map. Maps are produced using Google Maps (© Google Maps 2022) accessed using an API in R.

2015). A more targeted approach to simultaneously reduce congestion as well as emissions per vehicle per unit distance is therefore recommended to other cities looking to implement policies to tackle high traffic $NO_x$ emissions.

The observation that $NO_x$ emissions in central London during this continuing period of reduced mobility were thought to be dominated by heat and power generation is an important one. This is a transition which was expected to occur in the coming years but was brought forward in time by the pandemic, providing a glimpse into future air quality. As of 2020, there were 2659 CHP sites in the UK with additional widespread usage in Europe (Department for Business, Energy and Industrial Strategy (BEIS), 2021). Due to their increased efficiency and the push towards NetZero economies, they are expected to increase in popularity. Despite this period of drastically reduced transport emissions, all air quality monitoring sites (urban background, urban traffic and curbside) in London far exceeded the new WHO $NO_2$ air quality target. To achieve these targets it is therefore clear that legislation is required to reduce $NO_x$ emissions from heat and power generation. The heat and power generation source has been somewhat neglected due to the prominence of issues with diesel vehicle emissions. But with the planned use of hydrogen combustion in decarbonisation, which currently has major uncertainties due to a lack of experimental data, now is the critical time to start thinking about policy intervention for this sector (Lewis, 2021). This makes the lack of acknowledgement for gas combustion in boilers in the UK clean air strategy highly disappointing. This is the first indication from a megacity which shows heat and power emissions will need to be regulated to achieve the new air quality $NO_x$ targets. As more and more of the world's population is expected to live in urban areas, it is essential that compliance with WHO targets is achieved to minimise health and economic impacts. The conclusions derived from this work will therefore be of interest to other nations, especially with air quality improvements being increasingly sought in the developing world.

*Code availability.* The eddy4R v.0.2.0 software framework used to generate eddy-covariance flux estimates can be freely accessed at https://github.com/NEONScience/eddy4R. The eddy4R turbulence v0.0.16 software module was accessed under Terms of Use for this study (https://www.eol.ucar.edu/content/cheesehead-code-policy-appendix) and are available upon request.

*Data availability.* Automatic Urban and Rural Network data used in the analysis was obtained from https://uk-air.defra.gov.uk/networks/network-info?view=aurn under "Open Government Licence v3.0" (last access: August 2022). The London Atmospheric Emissions Inventory is available at https://naei.beis.gov.uk/ (© Crown 2022 copyright Defra & BEIS, licenced under the Open Government Licence (OGL)). For the measurement data in this paper, the calculated fluxes are not available in any repository due to the intensity of the processing and interpretation required. We are happy to make this available upon request. 15 min aggregated concentrations are available on the Centre for Environmental Data Analysis database, but these were not directly used here. The ERA5 boundary layer height data can be accessed at http://cds.climate.copernicus.eu/cdsapp#!/home (Copernicus Climate Change Service Climate Data Store (CDS), 2017). The traffic count data used in this article were provided by Transport for London (2021) (Automatic Traffic Counter data; original source data provided by Operational Analysis department, Transport for London).

*Author contributions.* SJC made the NO$_x$ measurements, calculated the fluxes and performed the analyses. SJC wrote the paper and produced the figures with input from co-authors. WSD provided support with the measurements, flux calculations and interpretation of the data. JDL provided support for the measurements and interpretation of the data. CH and EN provided supporting measurements from the site and

aided in interpretation of the data. SM provided training on the eddy4R software and aided in interpretation of the data. JFB provided meteorological data for the site.

*Competing interests.* The authors declare no competing interest.

*Acknowledgements.* Measurements were funded by the UK Natural Environment Research Council through the OSCA (Integrated Research Observation System for Clean Air) project of the Clean Air Strategic Priority Fund (Grant numbers NE/T001917/1 and NE/T001798/2)

and through award number NE/R016429/1 to UKCEH as part of the UK-SCAPE programme delivering National Capability. Samuel Cliff was supported by the Panorama Natural Environment Research Council (NERC) Doctoral Training Partnership (DTP), under grant NE/S007458/1. The National Ecological Observatory Network is a program sponsored by the National Science Foundation and operated under cooperative agreement by Battelle. This material is based in part upon work supported by the National Science Foundation through the NEON Program. The author also thanks Neil Mullinger and Karen Yeung (UK Center for Ecology and Hydrology) for instrument and

sample line maintenance, Ally Lewis and Rhianna Evans for help in manuscript preparation and British Telecom (BT) for granting use of the tall tower for research purposes, in particular Karen Ahern and Guille Parada for arranging work permits and facilitating access to the site.

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

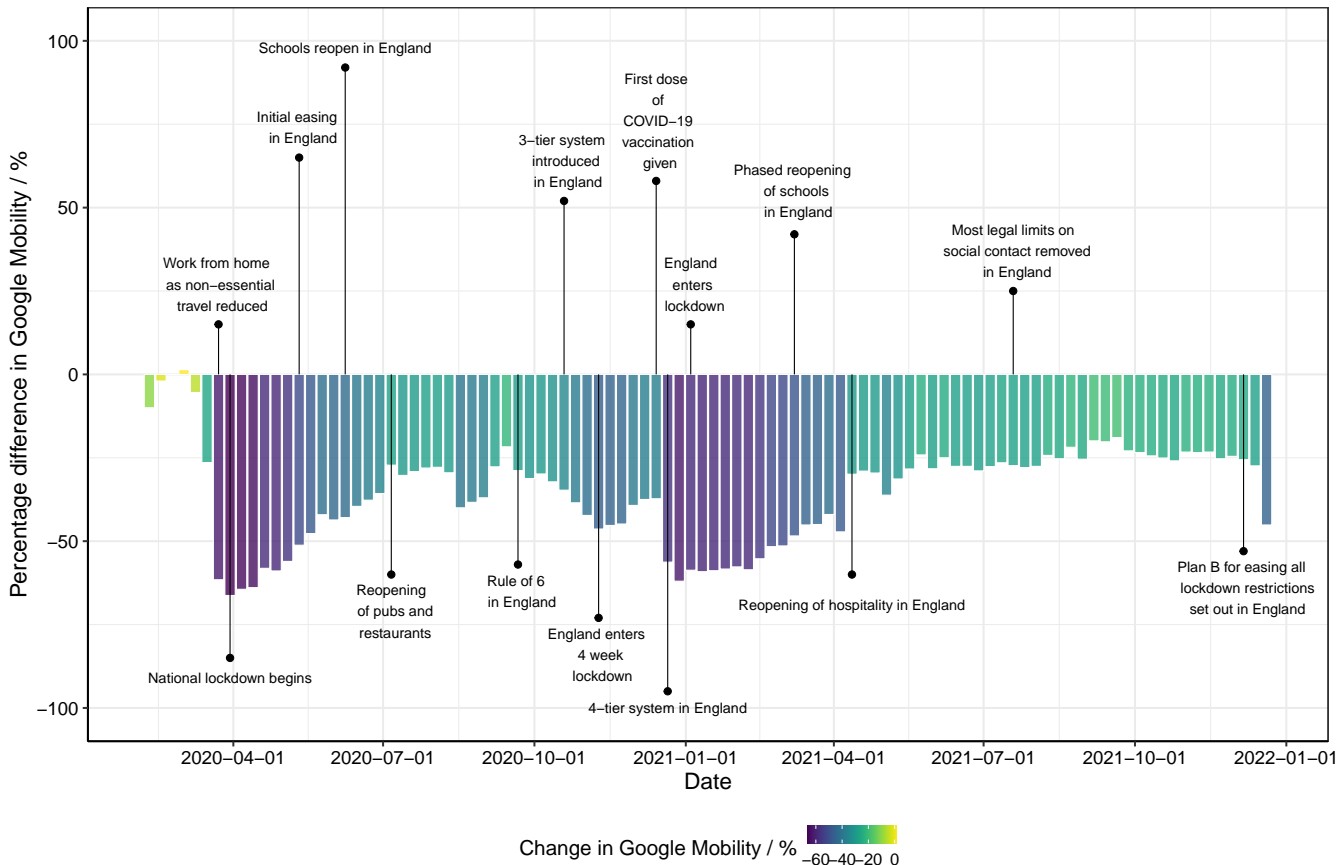

**Figure A1.** A timeline of COVID restrictions in England from the start of the COVID-19 pandemic until January 2022. Coloured bars represent the weekly change in Google Mobility across the UK.

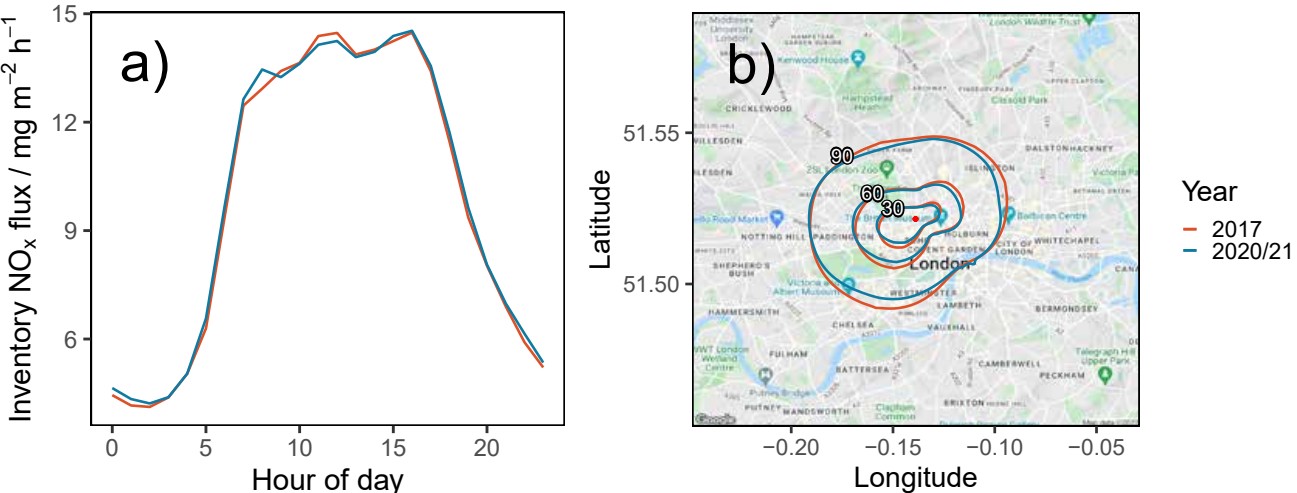

**Figure A2.** Comparison of a) the diurnal profile of the inventory generated time series and b) the 30, 60 and 90 % footprint contribution contours for the 2017 data set in black and the 2020/21 data set in red. Map in b) produced from Google Maps (© Google Maps 2021) accessed using an API in R.

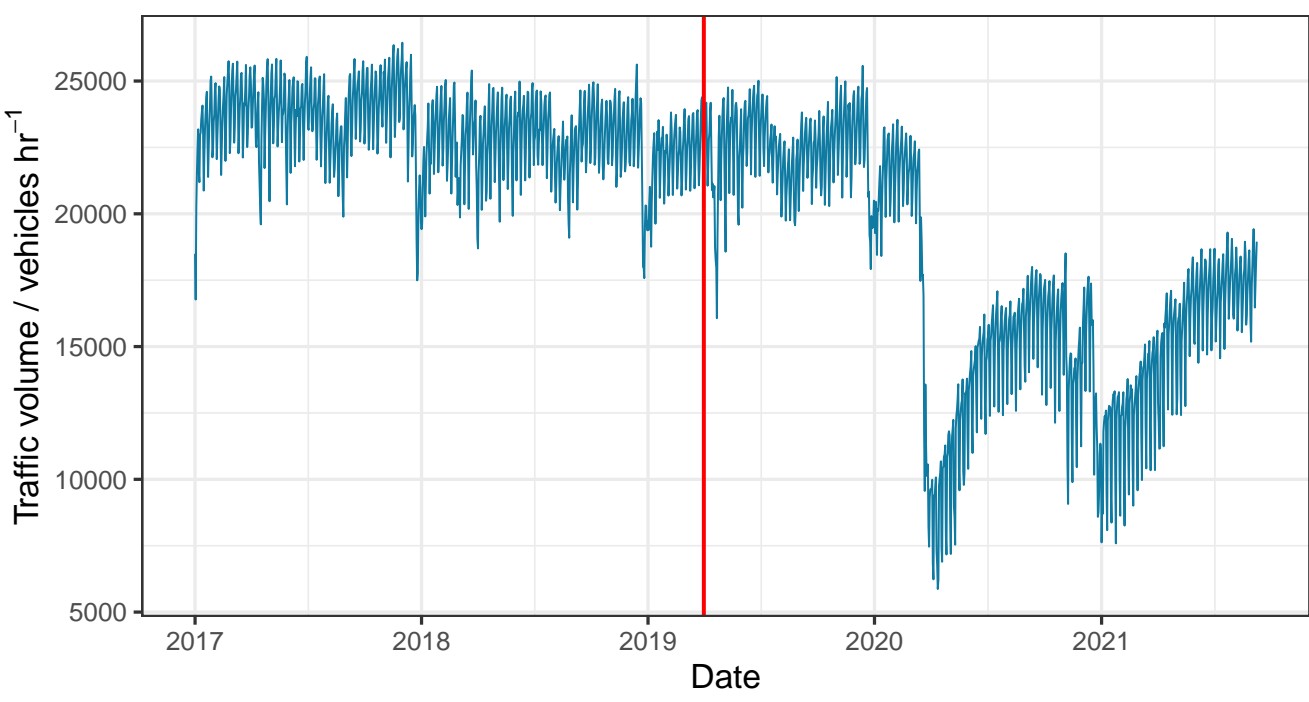

**Figure A3.** Daily average traffic load from 01/01/2017 - 01/09/2021. The date of the introduction of the ULEZ is marked as a vertical red line.

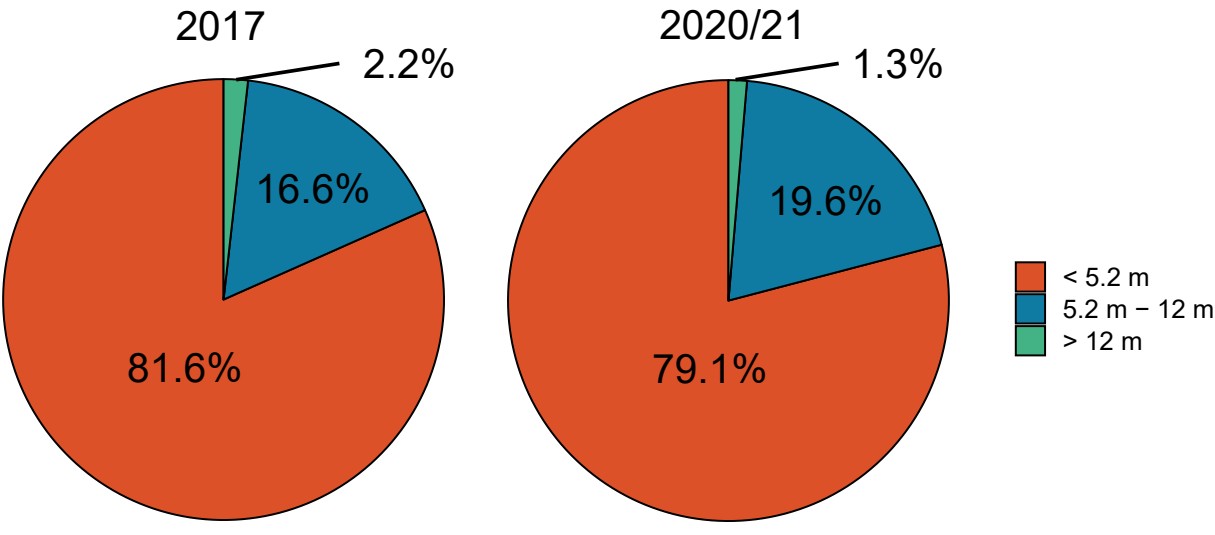

**Figure A4.** Pie chart for the 2017 and 2020/21 measurement periods displaying the distribution of vehicle length classes measured by the 24 ATC's surrounding the BT Tower.

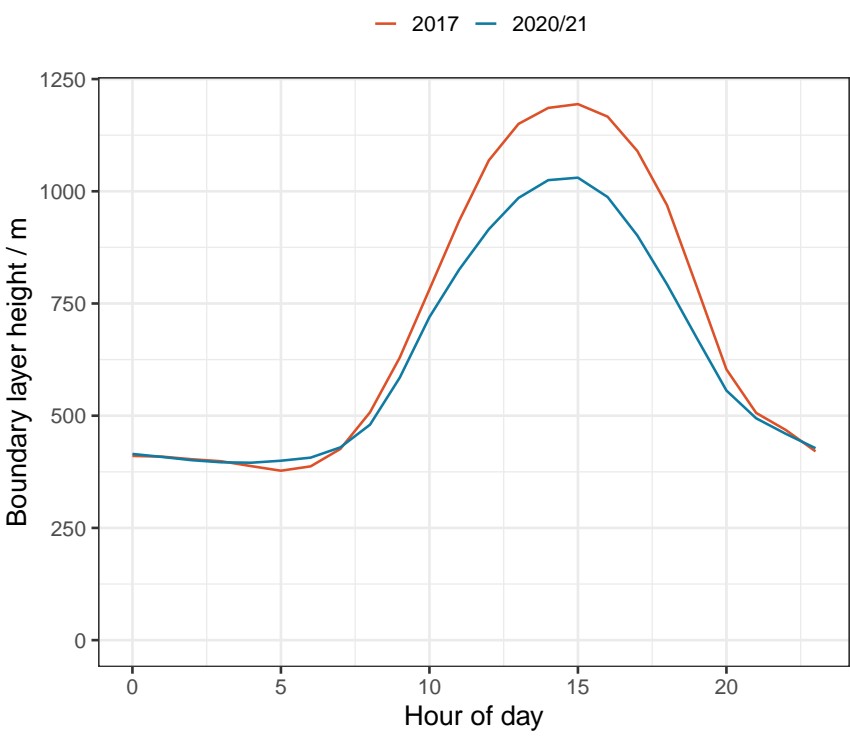

**Figure A5.** Boundary layer height comparison.

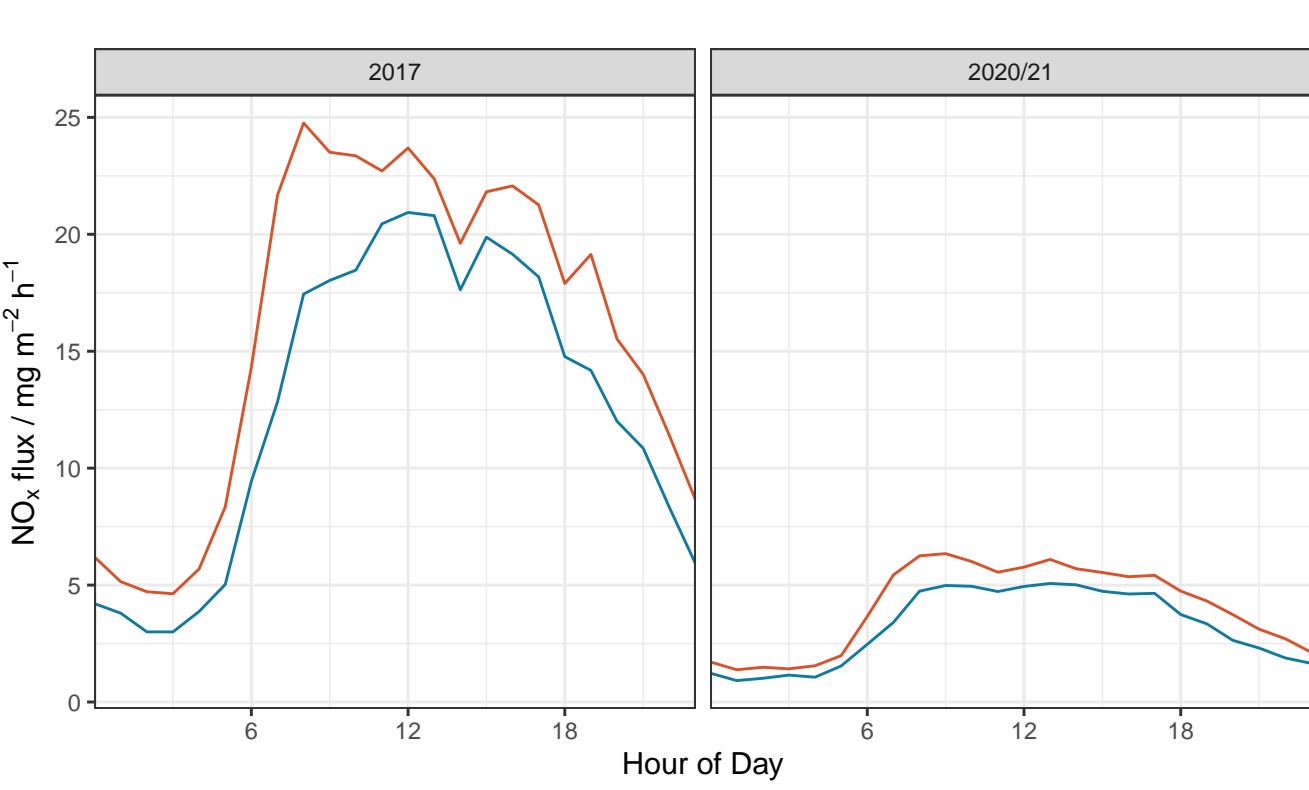

**Figure A6.** Diurnal profiles comparing uncorrected and vertical flux divergence corrected $NO_x$ flux for the 2017 and 2020/21 measurement periods.