# Peer review of "Pandemic Restrictions in 2020 highlight the significance of non-road $NO_x$ sources in central London"

_EGUsphere, 2022_

## Referee Comment (RC2)

This paper applied the eddy covariance technique to measure the NOx and CO2 flux in central London during the pandemic restriction period and by comparing it with the data obtained in 2017, authors evaluated the relative change of NOx flux, CO2 flux, and traffic load. With external constraints on NOx and CO2 emissions, the change in inventory sectors can be estimated. According to the spatial mapping analysis, the significant role of point sources on NOx emission was uncovered and authors recommended further legislation on heat and power generation to achieve the new WHO $NO_2$ air quality target. This paper is interesting by covering the urban NOx flux and its source investigation analysis. However, the urban-relevant flux data are quite limited because of the challenges of conducting flux measurements in the urban landscape. Overall, before considering further revision and potential publication, the comments below should be addressed.

1. Please provide more information to demonstrate such flux measurement setup on the BT tower fulfilled the requirement of the eddy covariance method. Several questions should be answered with the help of more detailed information including but not limited to:
whether the sampling height was within the inertial sublayer;
whether the data measuring frequency can cover the entire range of energy-carrying eddies;
whether the mast where the ultrasonic anemometer was attached was solid with little waggle;
whether the storage term and NOx chemistry had a significant impact on the measured flux…
2. It would be a rigorous approach to describe how lag-time was determined and what was the general QAQC results of the flux data according to the eddy4R software. The widely adopted 1-10 quality matrix is recommended to describe the quality control results instead of using high-quality.

Below are some more specific comments:
1. Line 29: Please add the reference for this sentence.
2. Line 61: The full name of BT tower should be added where it was first mentioned.
3. Line 127-128: Please define high-quality fluxes. Given the turbulent situation and characteristics of the city landscape, the flux data failed the QAQC criteria could be a lot based on my own experience. Therefore, specifying your QAQC flag matrix would be important,

4. Line 132-133: According to my reading of figure 2, the statement here was not accurate. The lowest traffic flow was in Jan. but clearly, the NOx flux during the same time was not the highest. Please improve the statement.

In terms of figure 2, I am quite interested in the trend of NOx flux from April to August. The traffic flow gradually increased as the stringency index decreased but the NOx flux decreased showing anti-correlation with traffic flow. This is odd to me. Maybe the authors can discuss this phenomenon.

5. Line 135-138: The comparison of the average diurnal profile between 2020/21 and 2017 data set cannot get the percentage reduction directly. I am guessing the 75% reduction of NOx flux referred to the difference in average NOx fluxes, then it would be clearer to include the actual value before the statement of the percentage change.

6. Line 156-157: Please add reference to the previous observations mentioned.

7. Line 161-163: There was another assumption that the emission characteristics of the heat and power generation remained the same so that the emission ratio of NOx and $CO_2$ was assumed to be constant. If there is any reference to support this assumption, it would be nice to have it cited.

8. Line 176: I might be wrong but I think, because of the modernization of the vehicle fleet resulting in lower NOx/$CO_2$ emission ratios, the hydrocarbons in the fuel can be more completely and efficiently converted to $CO_2$. If this is true, then the second bounding condition may not be the case. $CO_2$ emission can decrease by less than 32%.

9. Line 204: Figure 5 having the split data by wind direction was interesting. I also noticed that data points measured with east and north wind were less condense comparing the rest of the data. It looks like there were more data points or NOx emissions might come from sources that were less related to traffic flow. Maybe in the upwind footprint area of east and north, there were more heat and power generation sources? It would be great to include such a discussion.

---

## Author Response (AR1)

**Authors' Reponse to reviewer comments on egusphere-2022-956**

Please find below point-by-point responses to the reviewers' comments. You will find the exact comment from each reviewer in black italicised type followed by our response indented. Our response to the reviewer is in black type, and any changes to the text in blue. The line references for these quotations are those quoted by the reviewers and therefore those of the original manuscript.

Please note, additional edits have been made due to a mistake found in the QA/QC filtering of the 2020/21 flux data. These are outlined at the end of this document.

**Reviewer 1**

1.1 *This manuscript presents eddy covariance $NO_x$ and $CO_2$ fluxes measured in central London during the COVID pandemic and shows by comparing to pre-pandemic measurements that $NO_x$ emissions significantly decreased due to reductions in traffic load. While a number of studies have revealed $NO_x$ reductions during COVID from regional to global scales using satellite observations or surface monitors coupled with models, this study offers insight from another angle with the eddy covariance technique and delves into source attribution of $NO_x$ and $CO_2$ reductions. The study draws attention to urban power and heat generation, which was identified as the major source of $NO_x$ in the area during the lockdowns.*

> We would like to thank the reviewer for their time reading and reviewing our manuscript, and believe their comments have enabled us to improve the clarity of a number of things throughout the text.

1.2 *My overall comment is that the authors should discuss the extent to which their findings relate to and differentiate from existing studies in the field, and highlight the fact that this is the first evidence using eddy covariance measurements in a megacity. The only other eddy covariance measurements I am aware of that looked at this topic were made in Innsbruck, Austria (Lamprecht et al., 2021. doi: 10.5194/acp-21-3091-2021). Aside from this, I also have concerns regarding the method used for comparing the two periods and the conclusions drawn, mainly the lack of discussion on other factors that may vary between the two periods. Please see below the details.*

> Some text has been added discussing the Lamprecht et al. (2022) study and our comment in the introduction which mentions that these are the only $NO_x$ emissions measurements in a megacity has been expanded.

> "Recently, Lamprecht et al. (2021) have used long term air pollutant emissions measurements to understand how COVID-19 restrictions have impacted different sources of $NO_x$ in the small European city of Innsbruck, Austria. We undertake a similar analysis, but for the megacity of London. This offers the perspective of a different location where not only are emissions much higher, but contributions from different sources can vary significantly due to the nature of the activity required to support both greater population size and density. With the number of megacities consistently increasing, and expected to reach 43 in 2030 (up from 31 in 2018), improving our understanding of the air pollutant sources in them is as critical as ever (United Nations, 2019)."

1.3 *L65: Can you provide an estimate of lag time of your measurements for each species?*

> Yes, a sentence has been added to the text with the lag times of each species. An additional sentence has also been added in reply to Response 2.3 on the method for lag determination.

> "This resulted in median lag times of 7.2 s for NO, 7.6 s for $NO_2$ and 21 s for $CO_2$."

1.4 *L71: I understand that many technical details were described in Drysdale et al. (2022) for the 2017 measurements so only a summary is provided, but please at least cite the relevant work(s) here for those who would like to read further on the instruments and methodologies. For example, how exactly do you achieve "$NO_x$ free air"? What are the accuracies and precision's of your instruments?*

Some additional details have been added to the text and the relevant work cited.

"Both chemical species were measured using a dual channel chemiluminescence analyser (Air Quality Design Inc., Boulder Colorado, USA; 5 Hz) as described previously by Squires et al. (2020); Drysdale et al. (2022). The number of photons measured by the photomultiplier tube was converted into a part per trillion (ppt) mixing ratio using a five point calibration curve produced through dilutions of a 5 ppm NO in $N_2$ calibration standard (BOC Ltd., UK; traceable to the scale of the UK National Physical Laboratory, NPL) into $NO_x$ free air (generated from an external Sofnofil and activated charcoal trap)."

"TThe uncertainty of the NO measurement is given as $\pm$ 3%, resulting from uncertainties in the sample mass flow controller, calibration gas mass flow controller and calibration gas certification. The uncertainty for the $NO_2$ measurement is given as $\pm$ 4.7% due to the additional uncertainty in the conversion efficiency calculation, determined in the laboratory via variation in repeated tests. The precision for each channel is calculated as 53 ppt and 184 ppt for NO and $NO_2$ respectively from the standard deviation in all the hourly zeros conducted for the measurement period."

"Dry mass fractions ($1\sigma$ precision < 300 ppb) were measured initially using a cavity ringdown spectrometer (Model 1301-f, Picarro Inc., Santa Clara, California, USA; 10 Hz) as described by Helfter et al. (2016)."

1.5 *L87: The $NO_x$, $CO_2$, and meteorological measurements all have different sampling rates. How did you synchronize the measurements to calculate the hourly fluxes?*

The meteorological parameters and the chemical species are logged simultaneously in the same file so that no synchronisation is required. The lag time for each scalar quantity is then separately determined in the processing.

"For ease of processing, each chemical species is logged separately at it's maximum measurement frequency into a file with the sonic anemometer data averaged to the same measurement frequency."

1.6 *Please specify the sampling period of your 2017 measurements because this is an important detail. Based on my understanding the 2017 fluxes were only available from March to August, whereas your 2020/21 data covered a full year. Some of your comparisons between the two periods included only those in the same months (Figure 5) but the others compared the full year of 2020/21 to the six months of 2017 (Figure 2 and 5). How much bias would these comparisons of unequal lengths cause?*

A sentence has been added to highlight the sampling period of the 2017 measurements. All figures are comparing the full 2020/21 time period and the caption of Figure 5 is wrong. Thank you for pointing out this mistake. This has been corrected in the text. For potential bias between the measurement periods, please see Response 1.7 and 2.2.

"Long-term measurements of NO and $NO_2$ fluxes began in September 2020 with data presented here up to September 2021. Data is compared to previous measurements made by Drysdale et al. (2022) from March-August 2017."

"Comparison of the measured $NO_x$ flux with hourly traffic load (sum of the 24 surrounding ATCs) for March through August 2017 (red) and Sept 2020 - Sept 2021 (blue)."

1.7 *Besides, the potential influence of meteorology between different times of the year/between different years was never discussed. If your argument is that the emissions decreased due to anthropogenic reasons you need to prove that the meteorological effects were negligible or at least provide an uncertainty estimate. Can you show some meteorological data from your anemometer such as the average temperature diurnal profile for each of the two periods, or the average boundary layer height from ERA5?*

Boundary layer height was on average 8% lower in 2020/21 compared to the 2017 measurements (Fig. 1). This would impact the magnitude of the vertical flux divergence correction (see Response 2.2) which we do not apply in these measurements for reasons previously described Drysdale et al. (2022). However, the reduction percentages have been calculated for $NO_x$ between the two periods using the correction and this shows that it would have negligible impact ($< 1\%$) on this value. The other meteorological issue would be whether the measurement footprint was similar between the two periods, which has already been shown to be the case in Figure A2.

1.8 *In addition, did you compare the instrument performance in 2020/21 to 2017 to make sure the measurements were not affected by any system degradation such as long-term drifts?*

The instrument was calibrated regularly during both measurement periods. In doing so, any drift in performance is calibrated out.

1.9 *L133: Can you also mark the date of full removal of lockdown restrictions on Figure A3?*

The closest thing to full removal of lockdown restrictions during the timeline of this manuscipt's data was July 19th and is marked as 'Most legal limits on social contact removed in England'. Some small restrictions still remained in including isolation requirements if contacted by test and trace or if returning to the country internationally. Since the date for "full" removal of restrictions is out of the date range presented in this data, it has not been marked on the graph.

1.10 *L137: Can you describe in more detail how you calculated the reduction percentages from the diurnal profiles?*

Reduction percentages are calculated as follows:

$$\% \ reduction = \left(1 - \frac{2020 \ average}{2017 \ average}\right) \times 100 \tag{1}$$

Where the average for each year is taken from the diurnal profile data so as to remove any bias towards periods less affected by stationarity. Values for each year are have been added to the text to improve the clarity, as discussed in Response 2.9.

"Average diurnal $NO_x$ fluxes were down 73 % (3.45 vs 12.88 mg m$^{-2}$ h$^{-1}$). However, only a corresponding 20 % reduction in $CO_2$ flux (2455 vs 3062 mg m$^{-2}$ h$^{-1}$) and 32 % reduction in traffic load (16540 vs 24405 vehicles day$^{-1}$) around the measurement site was observed."

1.11 *L176: What is the rationale behind the assumption that $CO_2$ emissions reduction scales linearly with traffic load reduction?*

The assumption was made that there was **at least** a 32% reduction in $CO_2$ flux in line with a 32% reduction in traffic flow. The rationale behind this is that if you remove 32% of the vehicles from the roads then you would reduce $CO_2$ emissions by 32%. There are a number of factors which mean the actual reduction was likely higher than this. They include the reduction in congestion between the two periods, which leads to increased vehicle efficiency, and reduced emissions on average per vehiCle (see Response 2.12). As such, the assumption becomes at least 32%.

1.12 *Figure 2: Interesting that the $CO_2$ and $NO_x$ flux diurnal profiles both show a bimodal pattern peaking around noon and again around 3-4 pm. I thought the peaks would appear closer to the morning and evening rush hours, especially in the case of $NO_x$ given that Figure 3 suggests transport was the main source of $NO_x$ emissions. Can you explain why they display this pattern?*

The small peaks in the middle of the day are likely due to noise in the data and with the size of the uncertainty we do not think are particularly notable. There is not too much variation in traffic flow between rush hours and instead it remains high, as do the fluxes. The argument as to why the flux decreases before traffic flow in the evening is an interesting one. The two decreases are within around an hour of each other which is the same

length of our flux aggregation period. It may well be a result of a mismatch in whether each measurement is classed as that taken at the beginning or end of each hour. This could also be a result of boundary layer height dynamics in which storage (see Response 2.2) could shift rush hour peaks into different areas of the diurnal. We do, however, note that previous flux measurements at the BT Tower have been compared to those measured at a much lower rooftop site within the flux footprint, and both sites showed a very similar temporal pattern (Helfter et al. 2016).

1.13 *Also, are the error bars on the diurnal profiles the 1σ standard deviation of fluxes? Is the greater variability of fluxes in **2020/21** mainly due to the difference in temporal coverage?*

The error bars represent the average total error in the flux measurements, calculated as the addition in quadrature of the random and systematic errors as described by Mann and Lenschow (1994). A statement has been added to the figure caption containing this information. There is greater variability in the **2017** fluxes which is a consequence of the magnitude of the fluxes.

"Average median diurnal profiles with error bars (calculated as the combination of random and systematic errors in the flux calculations, as described by Mann and Lenschow (1994)) for the data are shown to the right in blue for 2020/21 in comparison to those generated from the 2017 data in red."

1.14 *Figure 5: While the differences between the 2020/21 and 2017 fluxes are noticeable, it is difficult to get a sense of how the data actually correlate with traffic flow from the figure. Can you calculate the correlation coefficients statistically? This will also aid your argument "The greatly reduced correlation with traffic load for the easterly 2020/21 data in Figure 5 is further evidence that the dominant source in this direction is heat and power generation." (L235-236)*

Spearman correlation coefficients for each year and wind direction have been calculated and added to the figure. There is reduced correlation in 2020/21 compared to 2017 as appeared at first glance. Crucially, however, there is much reduced correlation in the Easterly and Northerly directions where heat and power generation emissions are expected to be greatest. A discussion of this has been added to the text.

"The Spearman Correlation coefficients presented in Figure 5 give further evidence that the dominant source between the two periods has changed. Correlations between $NO_x$ flux and traffic load are reduced in 2020/21, in particular in the Easterly direction. Here, the lowest correlation is observed and high $NO_x$ fluxes are seen even at low traffic loads. These observations are in agreement with the spatial mapping interpretation in that heat and power generation is the dominant source from this direction."

1.15 *L20: has additional*

This has been amended as suggested.

1.16 *L44: other external stimuli*

This has been amended as suggested.

1.17 *L46: surface-atmosphere exchange*

This has been amended as suggested.

1.18 *L69: was converted into*

This has been amended as suggested.

1.19 *L113: artic lorries?*

This has been amended to "articulated".

1.20 *Figure 2: Consider moving this figure to a different place. It is currently placed awkwardly between the "Results and Discussion" section title and the first paragraph.*

This has been amended as suggested.

1.21 *Figure 4: Check equation labels on the figure: $CO_2$ Eq. (2) and $NO_x$ Eq. (3)?*

This has been amended as suggested.

**Reviewer 2**

2.1 *This paper applied the eddy covariance technique to measure the $NO_x$ and $CO_2$ flux in central London during the pandemic restriction period and by comparing it with the data obtained in 2017, authors evaluated the relative change of $NO_x$ flux, $CO_2$ flux, and traffic load. With external constraints on $NO_x$ and $CO_2$ emissions, the change in inventory sectors can be estimated. According to the spatial mapping analysis, the significant role of point sources on $NO_x$ emission was uncovered and authors recommended further legislation on heat and power generation to achieve the new WHO $NO_2$ air quality target. This paper is interesting by covering the urban $NO_x$ flux and its source investigation analysis. However, the urban relevant flux data are quite limited because of the challenges of conducting flux measurements in the urban landscape. Overall, before considering further revision and potential publication, the comments below should be addressed.*

We would like to thank the reviewer for their time and effort in reviewing the manuscript. We hope that the following explanations and additional details are satisfactory in addressing the concerns on the urban flux measurements.

2.2 *Please provide more information to demonstrate such flux measurement setup on the BT tower fulfilled the requirement of the eddy covariance method. Several questions should be answered with the help of more detailed information including but not limited to: whether the sampling height was within the inertial sublayer; whether the data measuring frequency can cover the entire range of energy-carrying eddies; whether the mast where the ultrasonic anemometer was attached was solid with little waggle; whether the storage term and $NO_x$ chemistry had a significant impact on the measured flux.*

The reviewer highlights some important points, most of which have been previously addressed in detail in the literature. Below, we collate this information and discuss each of these. In response, a section has been added to the manuscript titled 'Flux uncertainties' which contains much of the information outlined below.

The impact of $NO_x$ chemistry between emission and the measurement height has been shown to be minimal for typical transport times seen at the BT Tower. The major loss route of $NO_x$ to the atmosphere is due to the reaction between $NO_2$ and OH. The rate constant for this reaction for our location specific conditions can be calculated from (Jet Propulsion Laboratory, 2020):

$$k_f\left(T,[M]\right) = \left\{ \frac{k_\infty\left(T\right)k_0\left(T\right)[M]}{k_\infty\left(T\right)+k_0\left(T\right)[M]} \right\} 0.6^{\left\{1+\left[log_{10}\left(\frac{k_0(T)[M]}{k_\infty(T)}\right)\right]^2\right\}^{-1}} \tag{2}$$

Where:

$$k_0\left(T\right) = k_0^{298}\left(\frac{298}{T}\right)^n \tag{3}$$

$$k_\infty\left(T\right) = k_\infty^{298}\left(\frac{298}{T}\right)^m \tag{4}$$

$$[M] = \frac{PA_v}{RT} \tag{5}$$

Mean average values of pressure (989 hPa) and temperature (289 K) measured at the BT Tower are used to give $k_f(T,[M]) = 1.973 \times 10^{-11}$. Assuming a simple first order loss rate, the level of $NO_x$ loss to the atmosphere can then be estimated from:

$$\frac{[NO_x]}{[NO_x]_0} = e^{-k[OH]t} \tag{6}$$

Where a typical [OH] value for London is of the order of $2 \times 10^6$ (Lee et al. 2016) and $t$ is the transport time. Barlow et al. (2011) estimate a typical transport time of $< 10$ minutes for the BT Tower, although under stable conditions this could increase to 20-50 minutes. Inputting these values into Equation 6 gives a typical $NO_x$ loss of 2%, increasing up to 11% for a 50 minute transport time. The 11% loss represents the maximum loss observed at the tower, since it occurs under the most stable conditions during peak OH concentrations for Summer in London. In reality, this level of loss will not be observed in the data since stable conditions are filtered out in the QAQC process and the majority of the OH concentration present throughout the year is less than that observed at midday in the Summer. We therefore consider this to be a minor uncertainty.

Another valid concern and source of uncertainty is the size of the measurement height relative to the boundary layer. At 191 m, the sample inlet and sonic anemometer is often an appreciable portion of the boundary layer and can extend above the constant flux layer. On occasion, this results in concentration enhancements below the measurement height and an underestimation of the surface flux through vertical flux divergence. The impact of storage and vertical flux divergence at the BT Tower has been discussed previously by Helfter et al., 2016; Drysdale et al., 2022, and in the absence of concentration and wind measurements at different heights up the tower, remains to be a notable source of uncertainty in the measurement. Helfter et al. (2016) speculates that venting after the onset of turbulence would capture some, if not most of the material stored below the measurement height. Drysdale et al. (2022) demonstrates a correction for vertical flux divergence as a function of effective measurement height and effective entrainment height. The correction was typically around 20% for 2017, but is not applied to the data due to uncertainties in the boundary layer height data. Since this work studies relative magnitudes between two periods, the impact of VFD will likely cancel out, provided the meteorology is similar. Since boundary layer height was slightly lower on average (see Response 1.7) in 2020/21 compared to 2017, the % difference in $NO_x$ flux between the two periods was studied with the VFD correction applied (see Fig. 2). This was found to have minimal ($< 1\%$) impact and thus does not affect the story presented in this manuscript. We also note that similar levels of $NO_x$ fluxes have been observed during aircraft campaigns over London, in which the measurement height is even greater (Vaughan et al. 2021).

The sonic anemometer is located on a 12.2 m solid steel scaffolding tower at the top of the BT tower and is pictured in Lane et al. (2013). Instability in this set-up is very minimal and is not thought to be an issue for the flux measurements.

Measurements at 5 Hz or greater are thought to be sufficient for capturing the majority of the flux at the measurement location due to the large eddy size above the urban roughness layer. Drysdale et al. (2022) have calculated high frequency loss for NO and $NO_2$ at the BT Tower via cospectra relative to the temperature measured at 20 Hz. Correction factors above 1 Hz were shown to be of the order of 2-3 %. It is therefore considered a minor uncertainty. Similarly, low frequency loss is thought to be minimal. Previous studies at the BT Tower for 30-minute flux averaging periods have calculated losses due to high-pass filtering to be $< 5\%$ (Helfter et al. 2011; Langford et al., 2010b). Since a 60 minute averaging period is used here, loss will be even lower and as a result, no correction is applied.

"

**Flux uncertainties**

**$NO_x$ chemistry**

Eddy covariance has traditionally only been used for relatively unreactive greenhouse gases like $CO_2$ with long atmospheric lifetimes. Attempting the calculation of $NO_x$ fluxes is potentially problematic due to the greater reactivity and hence shorter lifetime of the species. If the loss rates of the reactive species is of a similar timescale to the vertical

[Figure]

Figure 1: Diurnal profiles comparing boundary layer height for the 2017 and 2020/21 measurement periods.

[Figure]

Figure 2: Diurnal profiles comparing uncorrected and vertical flux divergence corrected $NO_x$ flux for the 2017 and 2020/21 measurement periods.

transport to the measurement height, the measured flux would be an underestimate and would not be representative of those emitted at the ground. In the case of $NO_x$, the major loss route to the atmosphere is via the reaction between $NO_2$ and OH. The rate constant for this simple association reaction can be calculated for the BT Tower specific conditions from Eq. 7 using mean values of temperature (T, 289 K) and pressure (P, 989 hPa) (Jet Propulsion Laboratory, 2020). This is derived from the low-pressure limiting rate constant ($k_0(T)$) and the high-pressure limiting rate constant ($k_\infty(T)$) using location specific total gas concentrations ([M]). n and m are simple exponents for the given reaction, in this case 3 and 0 respectively.

$$k_f\left(T, [M]\right) = \left\{ \frac{k_\infty\left(T\right)k_0\left(T\right)[M]}{k_\infty\left(T\right) + k_0\left(T\right)[M]} \right\} 0.6^{\left\{ 1 + \left[ log_{10}\left( \frac{k_0(T)[M]}{k_\infty(T)} \right) \right]^2 \right\}^{-1}} \tag{7}$$

Where:

$$k_0\left(T\right) = k_0^{298}\left( \frac{298}{T} \right)^n \tag{8}$$

$$k_\infty\left(T\right) = k_\infty^{298}\left( \frac{298}{T} \right)^m \tag{9}$$

$$[M] = \frac{PA_v}{RT} \tag{10}$$

[revised manuscript text omitted]

2.3 *It would be a rigorous approach to describe how lag-time was determined and what was the general QAQC results of the flux data according to the eddy4R software. The widely adopted 1-10 quality matrix is recommended to describe the quality control results instead of using high-quality.*

A description of how the lag time is calculated has been added to the text. The QAQC in eddy4R is described in detail in a technical document by NEON (Metzger et al. 2022). Rather than the 1-10 quality matrix, data is flagged as either valid or invalid based on the combination of individual flags for data plausibility, homogeneity and stationarity, and development of turbulence. The word high-quality has been removed to avoid confusion with the 1-10 quality matrix.

"The lag time correction was determined by maximisation of the cross-covariance between the pollutant concentration and the vertical wind component with an additional application of a high-pass filter which improves the precision of the determined lag time by an order of magnitude (Hartmann et al., 2018; Squires et al., 2020)."

"The QA/QC process is described in detail by Smith and Metzger (2013); Metzger et al. (2022). Data is flagged as either valid or invalid based on the combination of individual flags for input data validation, homogeneity and stationarity, and development of turbulence."

2.4 *Line 29: Please add the reference for this sentence.*

The reference has been added as suggested.

2.5 *Line 61: The full name of BT tower should be added where it was first mentioned.*

The BT Tower acronym definition has been moved to the abstract where it was mentioned.

"Fluxes of nitrogen oxides ($NO_x$ = NO + $NO_2$) and carbon dioxide ($CO_2$) were measured using eddy covariance at the British Telecommunications (BT) Tower in central London during the coronavirus pandemic."

2.6 *Line 127-128: Please define high-quality fluxes. Given the turbulent situation and characteristics of the city landscape, the flux data failed the QAQC criteria could be a lot based on my own experience. Therefore, specifying your QAQC flag matrix would be important.*

You are correct in that the amount of urban flux data that fails QA/QC is a lot. A mistake was made (please see Author changes at the end of the document) in application of the flag to the 2020/21 $NO_x$ data and the correct number of hours passing QA/QC input to the text. Please see Response 2.3 for a discussion of the QA/QC matrix.

2.7 *Line 132-133: According to my reading of figure 2, the statement here was not accurate. The lowest traffic flow was in Jan. but clearly, the $NO_x$ flux during the same time was not the highest. Please improve the statement.*

The statement has been removed with a new discussion added in response to comment 2.8.

2.8 *In terms of figure 2, I am quite interested in the trend of $NO_x$ flux from April to August. The traffic flow gradually increased as the stringency index decreased but the $NO_x$ flux decreased showing anti-correlation with traffic flow. This is odd to me. Maybe the authors can discuss this phenomenon.*

I wonder whether this could be a result of reduced heat and power generation emissions due to the warmer weather.

"In fact, $NO_x$ flux displays an anti-correlation with traffic flow and stringency index from April to August. This is likely due to a reduction in heat and power generation emissions due to the warmer weather, which is a first indication that traffic may not be the dominant source of $NO_x$ flux during this period."

2.9 *Line 135-138: The comparison of the average diurnal profile between 2020/21 and 2017 data set cannot get the percentage reduction directly. I am guessing the 75% reduction of $NO_x$ flux referred to the difference in average $NO_x$ fluxes, then it would be clearer to include the actual value before the statement of the percentage change.*

Please see Response 1.10 for more details on how the percentage reductions were generated and improvements to the text. Values for each year have been added to the manuscript.

2.10 *Line 156-157: Please add reference to the previous observations mentioned.*

The references have been added.

"On the other hand, previous observations have shown a significant underestimation of $NO_x$ emissions in central London (Vaughan et al., 2021; Drysdale et al., 2022)."

2.11 *Line 161-163: There was another assumption that the emission characteristics of the heat and power generation remained the same so that the emission ratio of $NO_x$ and $CO_2$ was assumed to be constant. If there is any reference to support this assumption, it would be nice to have it cited.*

This assumption was based around the absence of any new legislation rather than a reference citing such information. However, the UK's Clean Air Strategy mentioned in the following sentence as part of the basis of the assumption has now been correctly referenced.

"With minimal legislation for the sector introduced between 2017 and 2020/21 and a failure to address $NO_x$ emissions from boilers in the UK's Clean Air Strategy, this assumption is considered reasonable (Department for Environment, Food and Rural Affairs (DEFRA), 2019)."

2.12 *Line 176: I might be wrong but I think, because of the modernization of the vehicle fleet resulting in lower $NO_x/CO_2$ emission ratios, the hydrocarbons in the fuel can be more completely and efficiently converted to $CO_2$. If this is true, then the second bounding condition may not be the case. $CO_2$ emission can decrease by less than 32%.*

The modernisation (including the transition to low/zero emitting vehicles) of the vehicle fleet actually results in a decrease in the average $CO_2$ emissions per new vehicle registration. The European Environment Agency reports a general downwards trajectory in $CO_2$ tailpipe emissions since 2000 for new vehicle registrations (https://www.eea.europa.eu/ims/co2-performance-of-new-passenger#ref-DNi82) which currently follows increasingly stringent emissions targets for $CO_2$. As such, our assumption that $CO_2$ emissions would have reduced by at least 32 % holds true.

"In reality, $CO_2$ emissions will have decreased by greater that 32 % as a result of the fleet modernisation which has lead to a decrease in the average $CO_2$ emissions per new vehicle registration (European Environment Agency, 2022)."

2.13 *Line 204: Figure 5 having the split data by wind direction was interesting. I also noticed that data points measured with east and north wind were less condense comparing the rest of the data. It looks like there were more data points or $NO_x$ emissions might come from sources that were less related to traffic flow. Maybe in the upwind footprint area of east and north, there were more heat and power generation sources? It would be great to include such a discussion.*

> Whilst there are fewer data points (owing to the common south westerly wind direction in London) the greater variability in the Northerly and Easterly fluxes is thought to be due to greater heat and power generation emissions which would not necessarily correlate well with traffic flow. This is mainly made up of Bloomsbury heat and power (already discussed in text) of which the majority of the emissions are from the east, but part of the UCL site does extend into the northerly quarter of Figure 5. A greater discussion of this has been included in Response 1.14

**Author made changes**

The availability of the eddy4R turbulence code in the "Code availability" section has been reworded.

"The eddy4R turbulence v0.0.16 software module was accessed under Terms of Use for this study (`https://www.eol.ucar.edu/content/cheesehead-code-policy-appendix`) and are available upon request."

An error was made in the QA/QC filtering for stationarity of the 2020/21 $NO_x$ flux data where the quality flag was not applied. Correctly applying the flag resulted in the removal of an extra 1978 hours of flux data. These values, presented on Line 126/127 have been corrected in the text as below.

"Of the 8760 hours in the year, 7034 hours of $NO_x$ fluxes were calculated. Data loss was largely due to instrument or sample pump failure. Of these 7034 hours, a further 3621 were removed by the quality control flagging to leave 3413 hours of $NO_x$ fluxes to be analysed."

This also lead to the recalculation of the % reduction for $NO_x$ between 2017 and 2020/21. The new value was found to be very similar, producing 73% compared to the previously stated 75%. As such, each mention of the value in the text has been altered from 75% to 73% as below.

Line 9, 137, 178, 181, Table 1 (Row 2 Column 2), Eq.3.

In addition, the congestion related factor discussed on line 270 has been altered accordingly from 22-47 % to 20 - 45 %.

Finally, the top facet of Figure 2, Figure 4, Figure 5 and Figure 6 have been replotted with the correct data. These changes have resulted in a, albeit minimal, appearance change.